# Exact Recovery of Mangled Clusters
# with Same-Cluster Queries

**Marco Bressan**\*
Dept. of CS, Univ. of Milan, Italy
marco.bressan@unimi.it

**Nicolò Cesa-Bianchi**
DSRC & Dept. of CS, Univ. of Milan, Italy
nicolo.cesa-bianchi@unimi.it

**Silvio Lattanzi**
Google
silviol@google.com

**Andrea Paudice**
Dept. of CS, Univ. of Milan, Italy &
Istituto Italiano di Tecnologia, Italy
andrea.paudice@unimi.it

## Abstract

We study the cluster recovery problem in the semi-supervised active clustering framework. Given a finite set of input points, and an oracle revealing whether any two points lie in the same cluster, our goal is to recover all clusters exactly using as few queries as possible. To this end, we relax the spherical $k$-means cluster assumption of Ashtiani et al. to allow for arbitrary ellipsoidal clusters with margin. This removes the assumption that the clustering is center-based (i.e., defined through an optimization problem), and includes all those cases where spherical clusters are individually transformed by any combination of rotations, axis scalings, and point deletions. We show that, even in this much more general setting, it is still possible to recover the latent clustering exactly using a number of queries that scales only logarithmically with the number of input points. More precisely, we design an algorithm that, given $n$ points to be partitioned into $k$ clusters, uses $\mathcal{O}(k^3 \ln k \ln n)$ oracle queries and $\widetilde{\mathcal{O}}(kn + k^3)$ time to recover the clustering with zero misclassification error. The $\mathcal{O}(\cdot)$ notation hides an exponential dependence on the dimensionality of the clusters, which we show to be necessary thus characterizing the query complexity of the problem. Our algorithm is simple, easy to implement, and can also learn the clusters using low-stretch separators, a class of ellipsoids with additional theoretical guarantees. Experiments on large synthetic datasets confirm that we can reconstruct clusterings exactly and efficiently.

## 1   Introduction

Clustering is a central problem of unsupervised learning with a wide range of applications in machine learning and data science. The goal of clustering is to partition a set of points in different groups, so that similar points are assigned to the same group and dissimilar points are assigned to different groups. A basic formulation is the $k$-clustering problem, in which the input points must be partitioned into $k$ disjoint subsets. A typical example is center-based $k$-clustering, where the points lie in a metric space and one is interested in recovering $k$ clusters that minimize the distance between the points and the cluster centers. Different variants of this problem, captured by the classic $k$-center, $k$-median, and $k$-means problems, have been extensively studied for several decades [1, 15, 26].

In this work we investigate the problem of recovering a latent clustering in the popular semi-supervised active clustering model of Ashtiani et al. [4]. In this model, we are given a set $X$ of $n$ input points in

$\mathbb{R}^d$ and access to an oracle. The oracle answers same-cluster queries (SCQs) with respect to a fixed but unknown $k$-clustering and tells whether any two given points in $X$ belong to the same cluster or not. The goal is to design efficient algorithms that recover the latent clustering while asking as few oracle queries as possible. Because SCQ queries are natural in crowd-sourcing systems, this model has been extensively studied both in theory [2, 3, 13, 19, 28, 29, 30, 33, 39] and in practice [12, 16, 37, 38] — see also [11] for other types of queries. In their work [4], Ashtiani et al. showed that by using $\mathcal{O}(\ln n)$ same-cluster queries one can recover the optimal $k$-means clustering of $X$ in polynomial time, whereas doing so without the queries would be computationally hard. Unfortunately, [4] relies crucially on a strong separation assumption, called $\gamma$-margin condition: for every cluster $C$ there must exist a sphere $S_C$, centered in the centroid $\mu_C$ of $C$, such that $C$ lies entirely inside $S_C$ and every point not in $C$ is at distance $(1 + \gamma)r_C$ from $\mu_C$, where $r_C$ is the radius of $S_C$. Thus, although [4] achieves cluster recovery with $\mathcal{O}(\ln n)$ queries, it does so only for a very narrow class of clusterings.

In this work we significantly enlarge the class of clusterings that can be efficiently recovered. We do so by relaxing the $\gamma$-margin condition of [4] in two ways (see Section 2 for a formal definition). First, we assume that every cluster $C$ has $\gamma$-margin in some latent space, obtained by linearly transforming all points according to some unknown positive semi-definite matrix $W_C$. This is equivalent to assume that $C$ is bounded by an ellipsoid (possibly degenerate) rather than by a sphere (which corresponds to $W_C = I$). This is useful because in many real-world applications the features are on different scales, and so each cluster tends to be distorted along specific directions causing ellipsoids to fit the data better than spheres [10, 22, 27, 31, 35]. Second, we allow the center of the ellipsoid to lie anywhere in space — in the centroid of $C$ or anywhere else, even outside the convex hull of $C$. This includes as special cases clusterings in the latent space which are solutions to $k$-medians, $k$-centers, or one of their variants. It is not hard to see that this setting captures much more general and challenging scenarios. For example, the latent clustering can be an optimal solution of $k$-centers where some points have been adversarially deleted and the features adversarially rescaled before the input points are handed to us. In fact, the latent clustering need not be the solution to an optimization problem, and in particular need not be center-based: it can be literally *any* clustering, as long as it respects the margin condition just described.

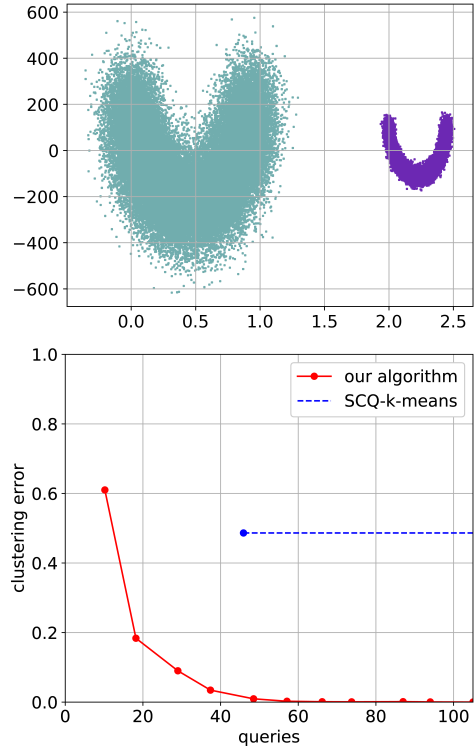

Figure 1: A toy instance on $10^5$ points that we solve exactly with 105 queries, while the scq-$k$-means algorithm of [4] is no better than random labeling.

Our main result is that, even in this significantly more general setting, it is still possible to recover the latent clustering *exactly*, in polynomial time, and using only $\mathcal{O}(\ln n)$ same-cluster queries. The price to pay for this generality is an exponential dependence of the number of queries on the dimension $d$ of the input space; this dependence is however unavoidable, as we show via rigorous lower bounds. Our algorithm is radically different from the one in [4], which we call scq-$k$-means here. The reason is that scq-$k$-means uses same-cluster queries to estimate the clusters' centroids and find their spherical boundaries via binary search. Under our more general setting, however, the clusters are not separated by spheres centered in their centroids, and thus scq-$k$-means fails, as shown in Figure 1 (see Section 8 for more experiments). Instead of binary search, we develop a geometric technique, based on careful tessellations of minimum-volume enclosing ellipsoids (MVEEs). The key idea is that MVEEs combine a low VC-dimension, which makes learning easy, with a small volume, which can be decomposed in easily classifiable elements. While MVEEs are not guaranteed to be consistent with the cluster samples, our results can be also proven using consistent ellipsoids that are close to the convex hull of the samples. This notion of low-stretch consistent ellipsoid is new, and may be interesting in its own right.

## 2 Preliminaries and definitions

All missing statements and proofs can be found in the supplementary material. The input to our problem is a triple $(X, k, \gamma)$ where $X \subset \mathbb{R}^d$ is a set of $n$ arbitrary points, $k \geq 2$ is an integer, and $\gamma \in \mathbb{R}_{>0}$ is the margin (see below). We assume there exists a latent clustering $\mathcal{C} = \{C_1, \dots, C_k\}$ over the input set $X$, which we do not know and want to compute. To this end, we are given access to an oracle answering *same-cluster queries*: a query $\mathrm{SCQ}(\boldsymbol{x}, \boldsymbol{x}')$ is answered by $+1$ if $\boldsymbol{x}, \boldsymbol{x}'$ are in the same cluster of $\mathcal{C}$, and by $-1$ otherwise. Our goal is to recover $\mathcal{C}$ while using as few queries as possible. Note that, given any subset $S \subseteq X$, with at most $k|S|$ queries one can always learn the label (cluster) of each $\boldsymbol{x} \in S$ up to a relabeling of $\mathcal{C}$, see [4].

It is immediate to see that if $\mathcal{C}$ is completely arbitrary, then no algorithm can reconstruct $\mathcal{C}$ with less than $n$ queries. Here, we assume some structure by requiring each cluster to satisfy a certain *margin* condition, as follows. Let $W \in \mathbb{R}^{d \times d}$ be some positive semidefinite matrix (possibly different for each cluster). Then $W$ induces the seminorm $\|\boldsymbol{x}\|_W = \sqrt{\boldsymbol{x}^\top W \boldsymbol{x}}$, which in turn induces the pseudo-metric $d_W(\boldsymbol{x}, \boldsymbol{y}) = \|\boldsymbol{x} - \boldsymbol{y}\|_W$. The same notation applies to any other PSD matrix, and when the matrix is clear from the context, we drop the subscript and write simply $d(\cdot, \cdot)$. The margin condition that we assume is the following:

**Definition 1** (Clustering margin)**.** *A cluster $C$ has margin $\gamma > 0$ if there exist a PSD matrix $W = W(C)$ and a point $\boldsymbol{c} \in \mathbb{R}^d$ such that for all $\boldsymbol{y} \notin C$ and all $\boldsymbol{x} \in C$ we have $d_W(\boldsymbol{y}, \boldsymbol{c}) > \sqrt{1 + \gamma}\, d_W(\boldsymbol{x}, \boldsymbol{c})$. If this holds for all clusters, then we say that the clustering $\mathcal{C}$ has margin $\gamma$.*

This is our only assumption. In particular, we do not assume the cluster sizes are balanced, or that $\mathcal{C}$ is the solution to an optimization problem, or that points in a cluster $C$ are closer to the center of $C$ than to the centers of other clusters. Note that the matrices $W$ and the points $\boldsymbol{c}$ are unknown to us. The spherical $k$-means setting of [4] corresponds to the special case where for every $C$ we have $W = rI$ for some $r = r(C) > 0$ and $\boldsymbol{c} = \boldsymbol{\mu}(C) = \frac{1}{|C|} \sum_{\boldsymbol{x} \in C} \boldsymbol{x}$.

We denote a clustering returned by our algorithm by $\widehat{\mathcal{C}} = \{\widehat{C}_1, \dots, \widehat{C}_k\}$. The quality of $\widehat{\mathcal{C}}$ is measured by the disagreement with $\mathcal{C}$ under the best possible relabeling of the clusters, that is, $\triangle(\widehat{\mathcal{C}}, \mathcal{C}) = \min_{\sigma \in S_k} \frac{1}{2n} \sum_{i=1}^k |C_1 \triangle \widehat{C}_{\sigma(i)}|$, where $S_k$ is the set of all permutations of $[k]$. Our goal is to minimize $\triangle(\widehat{\mathcal{C}}, \mathcal{C})$ using as few queries as possible. In particular, we characterize the query complexity of exact reconstruction, corresponding to $\triangle(\widehat{\mathcal{C}}, \mathcal{C}) = 0$. The *rank* of a cluster $C$, denoted by $\mathrm{rank}(C)$, is the rank of the subspace spanned by its points.

## 3 Our contribution

Our main contribution is an efficient active clustering algorithm, named RECUR, to recover the latent clustering exactly. We show the following.

**Theorem 1.** *Consider any instance $(X, k, \gamma)$ whose latent clustering $\mathcal{C}$ has margin $\gamma$. Let $n = |X|$, let $r \leq d$ be the maximum rank of a cluster in $\mathcal{C}$, and let $f(r, \gamma) = \max\left\{2^r, \mathcal{O}\left(\frac{r}{\gamma} \ln \frac{r}{\gamma}\right)^r\right\}$. Given $(X, k, \gamma)$, RECUR with probability 1 outputs $\mathcal{C}$ (up to a relabeling), and with high probability runs in time $\mathcal{O}((k \ln n)(n + k^2 \ln k))$ using $\mathcal{O}\big((k \ln n)(k^2 d^2 \ln k + f(r, \gamma))\big)$ same-cluster queries.*

More in general, RECUR clusters correctly $(1 - \varepsilon)n$ points using $\mathcal{O}\big((k \ln 1/\varepsilon)(k^2 d^2 \ln k + f(r, \gamma))\big)$ queries in expectation. Note that the query complexity depends on $r$ rather than on $d$, which is desirable as real-world data often exhibits a low rank (i.e., every point can be expressed as a linear combination of at most $r$ other points in the same cluster, for some $r \ll d$). In addition, unlike the algorithm of [4], which is Monte Carlo and thus can fail, RECUR is Las Vegas: it returns the correct clustering with probability 1, and the randomness is only over the number of queries and the running time. Moreover, RECUR is simple to understand and easy to implement. It works by recovering a constant fraction of some cluster at each round, as follows (see Section 5 and Section 6):

1. **Sampling.** We draw points uniformly at random from $X$ until, for some cluster $C$, we obtain a sample $S_C$ of size $\simeq d^2$. We can show that with good probability $|C| \simeq \frac{1}{k}|X|$, and that, by standard PAC bounds, any ellipsoid $E$ containing $S_C$ contains at least half of $C$.

2. **Computing the MVEE.** We compute the MVEE (minimum-volume enclosing ellipsoid) $E = E_{\mathrm{J}}(S_C)$ of $S_C$. As said, by PAC bounds, $E$ contains at least half of $C$. If we were lucky, $E$

would not contain any point from other clusters, and $E \cap X$ would be our large subset of $C$. Unfortunately, $E$ can contain an arbitrarily large number of points from $X \setminus C$. Our goal is to find them and recover $C \cap E$.

3. **Tessellating the MVEE.** To recover $C \cap E$, we partition $E$ into roughly $(d/\gamma)^d$ hyperrectangles, each one with the property of being monochromatic: its points are either all in $C$ or all in $X \setminus C$. Thanks to this special tessellation, with roughly $(d/\gamma)^d$ queries we can find all hyperrectangles containing only points of $C$, and thus compute $C \cap E$.

Our second contribution is to show a family of instances where every algorithm needs roughly $(1/\gamma)^r$ same-cluster queries to return the correct clustering. This holds even if the algorithm is allowed to fail with constant probability. Together with Theorem 1, this gives an approximate characterization of the query complexity of the problem as a function of $\gamma$ and $r$. That is, for ellipsoidal clusters, a margin of $\gamma$ is necessary and sufficient to achieve a query complexity that grows roughly as $(1/\gamma)^r$. This lower bound also implies that our algorithm is nearly optimal, even compared to algorithms that can fail. The result is given formally in Section 7.

Our final contribution is a set of experiments on large synthetic datasets. They show that our algorithm RECUR achieves exact cluster reconstruction efficiently, see Section 8.

## 4 Related work.

The semi-supervised active clustering (SSAC) framework was introduced in [4], together with the SCQ-$k$-means algorithm that recovers $\mathcal{C}$ using $O(k^2 \ln k + k \ln n)$ same-cluster queries. This is achieved via binary search under assumptions much stronger than ours (see above). In our setting, SCQ-$k$-means works only when every point $c$ is close to the cluster centroid and the condition number of $W$ is small (see the supplementary material); indeed, our experiments show that SCQ-$k$-means fails even when $W \simeq I$. Interestingly, even if binary search and its generalizations are at the core of many active learning techniques [32], here they do not seem to help. We remark that we depend on $\gamma$ in the same way as [4]: if $\gamma$ is a lower bound on the actual margin of $\mathcal{C}$, then the correctness is guaranteed, otherwise we may return any clustering. Clustering with same-cluster queries is also studied in [28], but they assume stochastic similarities between points that do not necessarily define a metric. Same-cluster queries for center-based clustering in metric spaces were also considered by [34], under $\alpha$-center proximity [5] instead of $\gamma$-margin (see [4, Appendix B] for a comparison between the two notions). Finally, [3] used same-cluster queries to obtain a PTAS for $k$-means. Unfortunately, this gives no guarantee on the clustering error: a good $k$-means value can be achieved by a clustering very different from the optimal one, and vice versa. From a more theoretical viewpoint, the problem has been extensively studied for clusters generated by a latent mixture of Gaussians [8, 20, 18].

As same-cluster queries can be used to label the points, one can also learn the clusters using standard pool-based active learning tools. For example, using quadratic feature expansion, our ellipsoidal clusters can be learned as hyperplanes. Unfortunately, the worst-case label complexity of actively learning hyperplanes with margin $\gamma < 1/2$ is still $\Omega\big((R/\gamma)^d\big)$, where $R$ is the radius of the smallest ball enclosing the points [14]. Some approaches that bypass this lower bound have been proposed. In [14] they prove an approximation result, showing that $\mathrm{OPT} \times \mathcal{O}\big(d \ln \frac{R}{\gamma}\big)$ queries are sufficient to learn any hyperplane with margin $\gamma$, where OPT is the number of queries made by the optimal active learning algorithm. Moreover, under distributional assumptions, linear separators can be learned efficiently with roughly $\mathcal{O}(d \ln n)$ label queries [6, 7, 9]. In a different line of work, [23] show that $\mathcal{O}\big((d \ln n) \ln \frac{R}{\gamma}\big)$ queries suffice for linear separators with margin $\gamma$ when the algorithm can also make comparison queries: for any two pairs of points $(\boldsymbol{x}, \boldsymbol{x}')$ and $(\boldsymbol{y}, \boldsymbol{y}')$ from $X$, a comparison query returns 1 iff $d_W(\boldsymbol{x}, \boldsymbol{x}') \leq d_W(\boldsymbol{y}, \boldsymbol{y}')$. As we show, comparison queries do not help learning the latent metric $d_W$ using metric learning techniques [25] (see the supplementary material). In general, the query complexity of pool-based active learning is characterized by the star dimension of the family of sets [17]. This implies that, if we allow for a non-zero probability of failure, then $\mathcal{O}(\mathfrak{s} \ln n)$ queries are sufficient for reconstructing a single cluster, where $\mathfrak{s}$ is the star dimension of ellipsoids with margin $\gamma$. To the best of our knowledge, this quantity is not known for ellipsoids with margin (not even for halfspaces with margin), and our results seem to suggest a value of order $(d/\gamma)^d$. If true, this would imply then the general algorithms of [17] could be used to solve our problem with a number of queries comparable to ours. However, note that our reconstructions are exact with probability one, and are achieved by simple algorithms that work well in practice.

# 5 Recovery of a single cluster with one-sided error

This section describes the core of our cluster recovery algorithm. The main idea is to show that, given any subset $S_C \subseteq C$ of some cluster $C$, if we compute a small ellipsoid $E$ containing $S_C$, then we can compute $C \cap E$ deterministically with a small number of queries.

Consider a subset $S_C \subseteq C$, and let $\mathrm{conv}(S_C)$ be its convex hull. The *minimum-volume enclosing ellipsoid* (MVEE) of $S_C$, also known as Löwner-John ellipsoid and denoted by $E_{\mathrm{J}}(S_C)$, is the volume-minimizing ellipsoid $E$ such that $S_C \subset E$ (see, e.g., [36]). The main result of this section is that $C \cap E_{\mathrm{J}}(S_C)$ is easy to learn. Formally, we prove:

**Theorem 2.** *Suppose we are given a subset $S_C \subseteq C$, where $C$ is any unknown cluster. Then we can learn $C \cap E_{\mathrm{J}}(S_C)$ using $\max\big\{2^r, \mathcal{O}\big(\frac{r}{\gamma}\ln\frac{r}{\gamma}\big)^r\big\}$ same-cluster queries, where $r = \mathrm{rank}(C)$ and $E_{\mathrm{J}}(S_C)$ is the minimum-volume enclosing ellipsoid of $S_C$.*

In the rest of the section we show how to learn $C \cap E_{\mathrm{J}}(S_C)$ and sketch the proof of the theorem.

**The MVEE.**    The first idea is to compute an ellipsoid $E$ that is "close" to $\mathrm{conv}(S_C)$. A $d$-rounding of $S_C$ is any ellipsoid $E$ satisfying the following (we assume the center of $E$ is the origin):

$$\frac{1}{d}E \subseteq \mathrm{conv}(S_C) \subseteq E \tag{1}$$

In particular, by a classical theorem by John [24], the MVEE $E_{\mathrm{J}}(S_C)$ is a $d$-rounding of $S_C$. We therefore let $E = E_{\mathrm{J}}(S_C)$. Note however that *any* $d$-rounding ellipsoid $E$ can be chosen instead, as the only property we exploit in our proofs is (1).

It should be noted that the ambient space dimensionality $d$ can be replaced by $r = \mathrm{rank}(S_C)$. To this end, before computing $E = E_{\mathrm{J}}(S_C)$, we compute the span $V$ of $S_C$ and a canonical basis for it using a standard algorithm (e.g., Gram-Schmidt). We then use $V$ as new ambient space, and search for $E_{\mathrm{J}}(S_C)$ in $V$. This works since $E_{\mathrm{J}}(S_C) \subset V$, and lowers the dimensionality from $d$ to $r \leq d$. From this point onward we still use $d$ in our notation, but all our constructions and claims hold unchanged if instead one uses $r$, coherently with the bounds of Theorem 2.

**The monochromatic tessellation.**    We now show that, by exploiting the $\gamma$-margin condition, we can learn $C \cap E_{\mathrm{J}}(S_C)$ with a small number of queries. We do so by discretizing $E_{\mathrm{J}}(S_C)$ into hyperrectangles so that, for each hyperrectangle, we need only one query to decide if it lies in $C$ or not. The crux is to show that there exists such a discretization, which we call *monochromatic tessellation*, consisting of relatively few hyperrectangles, roughly $(\frac{d}{\gamma}\ln\frac{d}{\gamma})^d$.

Let $E = E_{\mathrm{J}}(S_C)$. To describe the monochromatic tessellation, we first define the notion of monochromatic subset:

**Definition 2.** *A set $B \subset \mathbb{R}^d$ is* monochromatic *with respect to a cluster $C$ if it does not contain two points $\boldsymbol{x}, \boldsymbol{y}$ with $\boldsymbol{x} \in C$ and $\boldsymbol{y} \notin C$.*

Fix a hyperrectangle $R \subset \mathbb{R}^d$. The above definition implies that, if $B = R \cap E$ is monochromatic, then we learn the label of all points in $B$ with a single query. Indeed, if we take any $\boldsymbol{y} \in B$ and any $\boldsymbol{x} \in S_C$, the query $\mathrm{SCQ}(\boldsymbol{y}, \boldsymbol{x})$ tells us whether $\boldsymbol{y} \in C$ or $\boldsymbol{y} \notin C$ simultaneously for all $\boldsymbol{y} \in B$. Therefore, if we can cover $E$ with $m$ monochromatic hyperrectangles, then we can learn $C \cap E$ with $m$ queries. Our goal is to show that we can do so with $m \simeq (\frac{d}{\gamma}\ln\frac{d}{\gamma})^d$.

We now describe the construction in more detail; see also Figure 2. The first observation is that, if any two points $\boldsymbol{x}, \boldsymbol{y} \in X$ are such that $\boldsymbol{x} \in C$ and $\boldsymbol{y} \notin C$, then $|x_i - y_i| \gtrsim \gamma/d$ for some $i$. Indeed, if this was not the case then $\boldsymbol{x}, \boldsymbol{y}$ would be too close and would violate the $\gamma$-margin condition. This implies that, for $\rho \simeq 1 + \gamma/d$, any hyperrectangle whose sides have the form $[\beta_i, \beta_i\rho]$ is

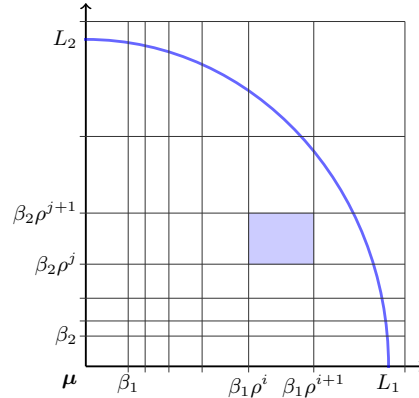

Figure 2: The tessellation $\mathcal{R}$ of $E \cap \mathbb{R}^d_+$. Every hyperrectangle $R$ (shaded) is such that $R \cap E$ is monochromatic, i.e. contains only points of $C$ or of $X \setminus C$.

monochromatic. We can exploit this observation to construct the tessellation. Let the semiaxes of $E$ be the canonical basis for $\mathbb{R}^d$ and its center $\boldsymbol{\mu}$ be the origin. For simplicity, we only consider the positive orthant, the argument being identical for every other orthant. Let $L_i$ be the length of the $i$-th semiaxis of $E$. The goal is to cover the interval $[0, L_i]$ along the $i$-th semiaxis of $E$ with roughly $\log_\rho(L_i/\beta_i)$ intervals of length increasing geometrically with $\rho$. More precisely, we let $T_i = \left\{ [0, \beta_i], (\beta_i, \beta_i\rho], \ldots, (\beta_i\rho^{b-1}, \beta_i\rho^b] \right\}$, where $\beta_i > 0$, $\rho > 1$, and $b \geq 0$ are functions of $\gamma$ and $d$. Then our tessellation is the cartesian product of all the $T_i$:

**Definition 3.** *Let $\mathbb{R}_+^d$ be the positive orthant of $\mathbb{R}^d$. The tessellation $\mathcal{R}$ of $E \cap \mathbb{R}_+^d$ is the set of $(b+1)^d$ hyperrectangles expressed in the canonical basis $\{\boldsymbol{u}_1, \ldots, \boldsymbol{u}_d\}$ of $E$: $\mathcal{R} = T_1 \times \ldots \times T_d$.*

We now come to the central fact. Loosely speaking, if $\beta_i \simeq \frac{\gamma}{d} L_i$ then the point $(\beta_1, \ldots, \beta_d)$ lies "well inside" $\mathrm{conv}(S_C)$, because (1) tells us $E$ itself is close to $\mathrm{conv}(S_C)$. By setting $\rho, b$ adequately, then, we can guarantee the intervals of $T_i$ of the form $(\beta_i\rho^{j-1}, \beta_i\rho^j]$ cover all the space between $\mathrm{conv}(S_C)$ and $E$. More formally we show that, for a suitable choice of $\beta_i, \rho, b$, the tessellation $\mathcal{R}$ satisfies the following properties (see the supplementary material):

(1) $|\mathcal{R}| \leq \max\left\{ 1, \mathcal{O}\!\left(\frac{d}{\gamma}\ln\frac{d}{\gamma}\right)^d \right\}$

(2) $E \cap \mathbb{R}_+^d \subseteq \bigcup_{R \in \mathcal{R}} R$

(3) For every $R \in \mathcal{R}$, the set $R \cap E$ is monochromatic w.r.t. $C$

Once the three properties are established, Theorem 2 immediately derives from the discussion above.

**Pseudocode.** We list below our algorithm that learns $C \cap E$ subject to the bounds of Theorem 2. We start by computing $E = E_J(S_C)$ and selecting the subset $E_X = X \cap E$. We then proceed with the tessellation, but without constructing $\mathcal{R}$ explicitly. Note indeed that, for every $\boldsymbol{y} \in E_X$, the hyperrectangle $R(\boldsymbol{y})$ containing $\boldsymbol{y}$ is determined uniquely by $|y_i|/\beta_i$ for all $i \in [d]$. In fact, we can manage all orthants at once by simply looking at $y_i/\beta_i$. After grouping all points $\boldsymbol{y}$ by their $R(\boldsymbol{y})$, we repeatedly take a yet-unlabeled $R$ and label it as $C$ or not $C$. Finally, we return all points in the hyperrectangles labeled as $C$.

---
**Algorithm 1** TessellationLearn$(X, S_C, \gamma)$
---
1: compute $E \leftarrow E_J(S_C)$ or any other $r$-rounding of $S_C$
2: compute $E_X \leftarrow X \cap E$
3: compute $\beta_i, \rho, b$ as a function of $r, \gamma$              ▷ see Figure 2
4: **for** every $\boldsymbol{y} \in E_X$ **do**
5:      map $\boldsymbol{y}$ to $R(\boldsymbol{y})$
6: $\boldsymbol{x}_C \leftarrow$ any point in $S_C$
7: **while** there is some unlabeled $R$ **do**
8:      $\mathrm{label}(R) \leftarrow \mathrm{SCQ}(\boldsymbol{x}_C, \boldsymbol{y})$, where $\boldsymbol{y}$ is any point s.t. $R(\boldsymbol{y}) = R$
9: **return** all $\boldsymbol{y}$ mapped to $R$ such that $\mathrm{label}(R) = +1$
---

**Low-stretch separators.** We conclude this section with a technical note. Although MVEEs enable exact cluster reconstruction, they do not give PAC guarantees since they do not ensure consistency. Indeed, if we draw a sample $S$ from $X$ and let $S_C = S \cap C$, there is no guarantee that $E = E_J(S_C)$ separates $S_C$ from $S \setminus S_C$. On the other hand, any ellipsoid $E$ separating $S_C$ from $S \setminus S_C$ is a good classifier in the PAC sense, but there is no guarantee it will be close to $\mathrm{conv}(S_C)$, thus breaking down our algorithm. Interestingly, in the supplementary material we show that it is possible to compute an ellipsoid that is simultaneously a good PAC classifier *and* close to $\mathrm{conv}(S_C)$, yielding essentially the same bounds as Theorem 2. Formally, we have:

**Definition 4.** *Given any finite set $X$ in $\mathbb{R}^d$ and a subset $S \subset X$, a $\Phi$-stretch separator for $S$ is any ellipsoid $E$ separating $S$ from $X \setminus S$ and such that $E \subseteq \Phi E_J(S)$.*

**Theorem 3.** *Suppose $C$ has margin $\gamma > 0$ w.r.t. to some $\boldsymbol{z} \in \mathbb{R}^d$ and fix any subset $S_C \subseteq C$. There exists a $\Phi$-stretch separator for $S_C$ with $\Phi = 64\sqrt{2}d^2 \max\left\{125, 1/\gamma^3\right\}$.*

# 6 Exact recovery of all clusters

In this section we conclude the construction of our algorithm RECUR (listed below), and we bound its query complexity and running time. RECUR proceeds in rounds. At each round, it draws samples uniformly at random from $X$ until, for some sufficiently large $b > 0$, it obtains a sample $S_C$ of size $bd^2 \ln k$ from some cluster $C$. At this point, by concentration and PAC bounds, we know that any ellipsoid $E$ containing $S_C$ satisfies $|C \cap E| \geq \frac{1}{4k}|X|$ with probability at least $1/2$. RECUR uses the routine TessellationLearn() from Section 5 to compute such a subset $C \cap E$ efficiently (see Theorem 2). RECUR then deletes $C \cap E$ from $X$ and repeats the process on the remaining points. This continues until a fraction $(1 - \varepsilon)$ of points have been clustered. In particular, when $\varepsilon < 1/n$, RECUR clusters all the points of $X$.

---
**Algorithm 2** RECUR$(X, k, \gamma, \varepsilon)$

---
1: $\widehat{C}_1, \ldots, \widehat{C}_k \leftarrow \emptyset$
2: **while** $|X| > \varepsilon n$ **do**
3:     draw samples with replacement from $X$ until $|S_C| \geq bd^2 \ln k$ for some $C$
4:     $C_E \leftarrow$ TessellationLearn$(X, S_C, \gamma)$
5:     add $C_E$ to the corresponding $\widehat{C}_i$
6:     $X \leftarrow X \setminus C_E$
7: **return** $\widehat{\mathcal{C}} = \{\widehat{C}_1, \ldots, \widehat{C}_k\}$

---

Regarding the correctness of RECUR, we have:

**Lemma 1.** *The clustering $\widehat{\mathcal{C}}$ returned by RECUR$(X, k, \gamma, \varepsilon)$ deterministically satisfies $\triangle(\widehat{\mathcal{C}}, \mathcal{C}) \leq \varepsilon$. In particular, for $\varepsilon < 1/n$ we have $\triangle(\widehat{\mathcal{C}}, \mathcal{C}) = 0$.*

This holds because $\triangle(\widehat{\mathcal{C}}, \mathcal{C})$ is bounded by the fraction of points that are still in $X$ when RECUR returns; and this fraction is at most $\varepsilon$ by construction. Regarding the cost of RECUR, we have:

**Lemma 2.** RECUR$(X, k, \gamma, \varepsilon)$ *makes* $\mathcal{O}(k^3 \ln k \ln(1/\varepsilon))$ *same-cluster queries in expectation, and for all fixed $a \geq 1$,* RECUR$(X, k, \gamma, 0)$ *with probability at least $1 - n^{-a}$ makes $\mathcal{O}(k^3 \ln k \ln n)$ same-cluster queries and runs in time $\mathcal{O}((k \ln n)(n + k^2 \ln k)) = \widetilde{\mathcal{O}}(kn + k^3)$.*

In the rest of the section we sketch the proof of Lemma 2. We start by bounding the number of rounds performed by RECUR. Recall that, at each round, with probability at least $1/2$ a fraction at least $1/4k$ of points are labeled and removed. Thus, at each round, the size of $X$ drops by $(1 - 1/8k)$ in expectation. Hence, we need roughly $8k \ln(1/\varepsilon)$ rounds before the size of $X$ drops below $\varepsilon n$. Indeed, we prove:

**Lemma 3.** RECUR*$(X, k, \gamma, \varepsilon)$ makes at most $8k \ln(1/\varepsilon)$ rounds in expectation, and for all fixed $a \geq 1$,* RECUR*$(X, k, \gamma, 0)$ with probability at least $1 - n^{-a}$ performs at most $(8k + 6a\sqrt{k}) \ln n$ rounds.*

We can now bound the query cost and running time of RECUR, by counting the work done at each round and using Lemma 3. To simplify the discussion we treat $d, r, \gamma$ as constants, but fine-grained bounds can be derived immediately from the discussion itself.

**Query cost of** RECUR**.** The algorithm makes queries at line 3 and line 4. At line 3, RECUR draws at most $bkd^2 \ln k = \mathcal{O}(k \ln k)$ samples. This holds since there are at most $k$ clusters, so after $bkd^2 \ln k$ samples, the condition $|S_C| \geq bd^2 \ln k$ will hold for some $C$. Since learning the label of each sample requires at most $k$ queries, line 3 makes $\mathcal{O}(k^2 \ln k)$ queries in total. At line 4, RECUR makes $f(d, \gamma) = \mathcal{O}(1)$ queries by Theorem 2. Together with Lemma 3, this implies that RECUR with probability at least $1 - n^{-a}$ makes at most $\mathcal{O}(k \ln n) \times \mathcal{O}(k^2 \ln k) = \mathcal{O}(k^3 \ln k \ln n)$ queries.

**Running time of** RECUR**.** Line 3 takes time $\mathcal{O}(k^2 \ln k)$, see above. The rest of each round is dominated by the invocation of TessellationLearn at line 4. Recall then the pseudocode of TessellationLearn from Section 5. At line 1, computing $E = E_J(S_C)$ or any $r$-rounding of $S_C$ takes time $\mathcal{O}(|S_C|^{3.5} \ln |S_C|)$, see [24].[2] This is in $\widetilde{\mathcal{O}}(1)$ since by construction $|S_C| = \mathcal{O}(d^2 \ln k) = \widetilde{\mathcal{O}}(1)$. Computing $E_X = X \cap E$ takes time $\mathcal{O}(|X| \text{poly}(d)) = \mathcal{O}(n)$. For the index (line 4), we can build in time $\mathcal{O}(|X \cap E|)$ a dictionary that maps every $R \in \mathcal{R}$ to the set $R \cap E_X$. The classification part

(line 7) takes time $|\mathcal{R}| = \mathcal{O}(1)$. Finally, enumerating all positive $R$ and concatenating the list of their points takes again time $\mathcal{O}(|X \cap E| \operatorname{poly}(d))$. By the rounds bound of Lemma 3, then, RECUR with probability at least $1 - n^{-a}$ runs in time $\mathcal{O}((k \ln n)(n + k^2 \ln k))$.

# 7 Lower bounds

We show that any algorithm achieving exact cluster reconstruction must, in the worst case, perform a number of same-cluster queries that is exponential in $d$ (the well-known "curse of dimensionality"). Formally, in the supplementary material we prove:

**Theorem 4.** *Choose any possibly randomized learning algorithm. There exist:*

1. *for all $\gamma \in (0, 1/7)$ and $d \geq 2$, an instance on $n = \Omega\big(\big(\frac{1+\gamma}{8\gamma}\big)^{\frac{d-1}{2}}\big)$ points and 3 clusters*

2. *for all $\gamma > 0$ and $d \geq 48(1+\gamma)^2$, an instance on $n = \Omega\big(e^{\frac{d}{48(1+\gamma)^2}}\big)$ points and 2 clusters*

*such that (i) the latent clustering $\mathcal{C}$ has margin $\gamma$, and (ii) to return with probability $2/3$ a $\widehat{\mathcal{C}}$ such that $\triangle(\widehat{\mathcal{C}}, \mathcal{C}) = 0$, the algorithm must make $\Omega(n)$ same-cluster queries in expectation.*

The lower bound uses two different constructions, each one giving a specific instance distribution where any algorithm must perform $\Omega(n)$ queries in expectation, where $n$ is exponential in $d$ as in the statement of the theorem. The first construction is similar to the one shown in [14]. The input set $X$ is a packing of $\simeq (1/\gamma)^d$ points on the $d$-dimensional sphere, at distance $\simeq \sqrt{\gamma}$ from each other. We show that, for $\boldsymbol{x} = (x_1, \ldots, x_d) \in X$ drawn uniformly at random, setting $W = (1 + \gamma) \operatorname{diag}(x_1^2, \ldots, x_d^2)$ makes $\boldsymbol{x}$ an outlier. That is, $X \setminus \{\boldsymbol{x}\}$ forms a first cluster $C_1$, and $\{\boldsymbol{x}\}$ forms a second cluster $C_2$, and both clusters satisfy the margin condition. In order to output the correct clustering, any algorithm must find $\boldsymbol{x}$, which requires $\Omega(n)$ queries in expectation. In the second construction, $X$ is a random sample of $n \simeq \exp(d/(1+\gamma)^2)$ points from the $d$-dimensional hypercube $\{0, 1\}^d$ such that each coordinate is independently 1 with probability $\simeq \frac{1}{1+\gamma}$. Similarly to the first construction we show that, for $\boldsymbol{x} \in X$ drawn uniformly at random, setting $W = (1 + \gamma) \operatorname{diag}(x_1, \ldots, x_d)$ makes $\boldsymbol{x}$ an outlier, and any algorithm needs $\Omega(n)$ queries to find it.

# 8 Experiments

We implemented our algorithm RECUR and compared it against SCQ-$k$-means [4]. To this end, we generated four synthetic instances on $n = 10^5$ points with increasing dimension $d = 2, 4, 6, 8$. The latent clusterings consist of $k = 5$ ellipsoidal clusters of equal size, each one with margin $\gamma = 1$ w.r.t. a random center and a random PSD matrix with condition number $\kappa = 100$, making each cluster stretched by $10\times$ in a random direction. To account for an imperfect knowledge of the data, we fed RECUR with a value of $\gamma = 10$ (thus, it could in principle output a wrong clustering). We also adopted for RECUR the batch sampling of SCQ-$k$-means, i.e., we draw $k \cdot 10$ samples in each round; this makes RECUR slightly less efficient than with its original sampling scheme (see line 3).

To further improve the performance of RECUR, we use a simple "greedy hull expansion" heuristic that can increase the number of points recovered at each round without performing additional queries. Immediately after taking the sample $S_C$, we repeatedly expand its convex hull $\operatorname{conv}(S_C)$ by a factor $\simeq (1 + \gamma/d)$, and add all the points that fall inside it to $S_C$. If $C$ is sufficiently dense, a substantial fraction of it will be added to $S_C$; while, by the margin assumption, no point outside $C$ will ever be added to $S_C$ (see the proof of the tessellation). This greedy hull expansion is repeated until no new points are found, in which case we proceed to compute the MVEE and the tessellation.

Figure 3 shows for both algorithms the clustering error $\triangle$ versus the number of queries, round by round, averaged over 10 independent runs (SCQ-$k$-means has a single measurement since it runs "in one shot"). The run variance is negligible and we do not report it. Observe that the error of SCQ-$k$-means is always in the range 20%–40%. In contrast, the error of RECUR decreases exponentially with the rounds until the latent clustering is exactly recovered, as predicted by our theoretical results. To achieve $\triangle \leq .05$, RECUR uses less than 3% of the queries needed by a brute force labeling, which is $kn = 5 \times 10^5$. Note that, except when clusters are aligned as in Figure 1, SCQ-$k$-means continues to perform poorly even after whitening the input data to compensate for skewness. Finally, note how the number of queries issued by RECUR increases with the dimensionality $d$, in line with Theorem 4.

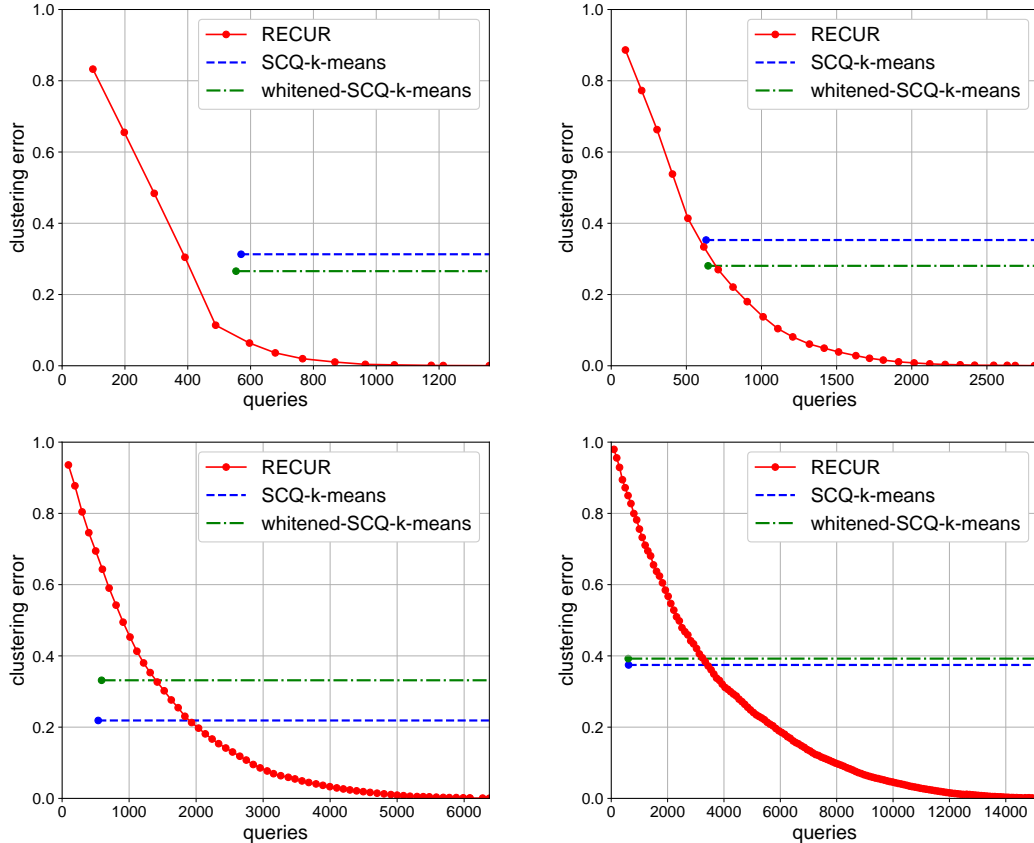

Figure 3: Clustering error vs. number of queries for $k = 5$ and $d = 2, 4, 6, 8$ (left to right, top to bottom). While SCQ-$k$-means performs rather poorly, RECUR always achieves exact reconstruction.

# 9 Conclusions

We have given a novel technique that, under general conditions, allows one to actively recover a clustering using only $\mathcal{O}(\ln n)$ same-cluster queries. Unlike previous work, our technique is robust to distortions and manglings of the clusters, and works for arbitrary clusterings rather than only for those based on the solution of an optimization problem. Our work leaves open three main questions:

**Q1:** Can our assumptions be strengthened in order to reduce the dependence on the dimension from exponential to polynomial, but without falling back to the setting of Ashtiani et al. [4]?

**Q2:** Can our assumptions be further relaxed, for instance by assuming a class of transformations more general than those given by PSD matrices?

**Q3:** Is there a natural and complete characterization of the class of clusterings that can be reconstructed with $\mathcal{O}(\ln n)$ queries?

## Acknowledgements

The authors gratefully acknowledge partial support by the Google Focused Award "Algorithms and Learning for AI" (ALL4AI). Marco Bressan was also supported in part by the ERC Starting Grant DMAP 680153, by the "Dipartimenti di Eccellenza 2018-2022" grant awarded to the Department of Computer Science of the Sapienza University of Rome, and by BICI, the Bertinoro International Center for Informatics. Nicolò Cesa-Bianchi is also supported by the MIUR PRIN grant Algorithms, Games, and Digital Markets (ALGADIMAR) and by the EU Horizon 2020 ICT-48 research and innovation action under grant agreement 951847, project ELISE (European Learning and Intelligent Systems Excellence).

## Broader impact

This work does not present any foreseeable societal consequence.

## Footnotes

\*Most of this work was done while the author was at the Sapienza University of Rome.

[2]More precisely, for a set $S$ an ellipsoid $E$ such that $\frac{1}{(1+\varepsilon)d} E \subset \text{conv}(S) \subset E$ can be computed in $\mathcal{O}(|S|^{3.5} \ln(|S|/\varepsilon))$ operations in the real number model of computation, see [24].

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
