[Supplementary Material]

# Exact Recovery of Mangled Clusters with Same-Cluster Queries (supplementary material)

**Marco Bressan**
Dept. of CS, Univ. of Milan, Italy
marco.bressan@unimi.it

**Nicolò Cesa-Bianchi**
DSRC & Dept. of CS, Univ. of Milan, Italy
nicolo.cesa-bianchi@unimi.it

**Silvio Lattanzi**
Google
silviol@google.com

**Andrea Paudice**
Dept. of CS, Univ. of Milan, Italy &
Istituto Italiano di Tecnologia, Italy
andrea.paudice@unimi.it

## 1 Ancillary results

### 1.1 VC-dimension of ellipsoids

For any PSD matrix $M$, we denote by $E_M = \{\boldsymbol{x} \in \mathbb{R}^d : d_M(\boldsymbol{x}, \boldsymbol{\mu}) \leq 1\}$ the $\boldsymbol{\mu}$-centered ellipsoid with semiaxes of length $\lambda_1^{-1/2}, \ldots, \lambda_d^{-1/2}$, where $\lambda_1, \ldots, \lambda_d \geq 0$ are the eigenvalues of $M$. We recall the following classical VC-dimension bound (see, e.g., [3]).

**Theorem 5.** *The VC-dimension of the class $\mathcal{H} = \{E_M : M \in \mathbb{R}^d, M \succeq 0\}$ of (possibly degenerate) ellipsoids in $\mathbb{R}^d$ is $\frac{d^2+3d}{2}$.*

### 1.2 Generalization error bounds

The next result is a simple adaptation of the classical VC bound for the realizable case (see, e.g., [5, Theorem 6.8]).

**Theorem 6.** *There exists a universal constant $c > 0$ such that for any family $\mathcal{H}$ of measurable sets $E \subset \mathbb{R}^d$ of VC-dimension $d < \infty$, any probability distribution $\mathcal{D}$ on $\mathbb{R}^d$, and any $\varepsilon, \delta \in (0, 1)$, if $S$ is a sample of $m \geq c\frac{d\ln(1/\varepsilon)+\ln(1/\delta)}{\varepsilon}$ points drawn i.i.d. from $\mathcal{D}$, then for any $E^* \in \mathcal{H}$ we have:*

$$\mathcal{D}(E \triangle E^*) \leq \varepsilon \qquad \text{and} \qquad \mathcal{D}(E' \setminus E^*) \leq \varepsilon$$

*with probability at least $1 - \delta$ with respect to the random draw of $S$, where $E$ is any element of $\mathcal{H}$ such that $E \cap S = E^* \cap S$, and $E'$ is any element of $\mathcal{H}$ such that $E^* \cap S \subseteq E' \cap S$.*

The first inequality is the classical PAC bound for the zero-one loss, which uses the fact that the VC dimension of $\{E \triangle E^* : E \in \mathcal{H}\}$ is the same as the VC dimension of $\mathcal{H}$. The second inequality follows immediately from the same proof by noting that, for any $E^* \in \mathcal{H}$ the VC dimension of $\{E \setminus E^* : E \in \mathcal{H}\}$ is not larger than the VC dimension of $\mathcal{H}$ because, for any sample $S$ and for any $F, G \in \mathcal{H}$, $(F \setminus E^*) \cap S \neq (G \setminus E^*) \cap S$ implies $F \cap S \neq G \cap S$.

### 1.3 Concentration bounds

We recall standard concentration bounds for non-positively correlated binary random variables, see [2]. Let $X_1, \ldots, X_n$ be binary random variables. We say that $X_1, \ldots, X_n$ are non-positively

correlated if for all $I \subseteq \{1, \ldots, n\}$ we have:

$$\mathbb{P}\big(\forall i \in I : X_i = 0\big) \leq \prod_{i \in I} \mathbb{P}(X_i = 0) \quad \text{and} \quad \mathbb{P}\big(\forall i \in I : X_i = 1\big) \leq \prod_{i \in I} \mathbb{P}(X_i = 1) \quad (1)$$

**Lemma 4** (Chernoff bounds). *Let $X_1, \ldots, X_n$ be non-positively correlated binary random variables. Let $a_1, \ldots, a_n \in [0, 1]$ and $X = \sum_{i=1}^{n} a_i X_i$. Then, for any $\varepsilon > 0$, we have:*

$$\mathbb{P}\big(X < (1 - \varepsilon)\mathbb{E}[X]\big) < e^{-\frac{\varepsilon^2}{2}\mathbb{E}[X]} \tag{2}$$

$$\mathbb{P}\big(X > (1 + \varepsilon)\mathbb{E}[X]\big) < e^{-\frac{\varepsilon^2}{2+\varepsilon}\mathbb{E}[X]} \tag{3}$$

### 1.4 Yao's minimax principle

We recall Yao's minimax principle for Monte Carlo algorithms. Let $\mathcal{A}$ be a finite family of deterministic algorithms and $\mathcal{I}$ a finite family of problem instances. Fix any two distributions $\boldsymbol{p}$ over $\mathcal{I}$ and $\boldsymbol{q}$ over $\mathcal{A}$, and any $\delta \in [0, 1/2]$. Let $\min_{A \in \mathcal{A}} \mathbb{E}_{I \sim \boldsymbol{p}}[C_\delta(I, A)]$ be the minimum, over every algorithm $A$ that fails with probability at most $\delta$ over the input distribution $\boldsymbol{p}$, of the expect cost of $A$ over the input distribution itself. Similarly, let $\max_{I \in \mathcal{I}} \mathbb{E}_{A \sim \boldsymbol{q}}[C_\delta(I, A)]$ be the expected cost of the randomized algorithm defined by $\boldsymbol{q}$ under its worst input from $\mathcal{I}$, assuming it fails with probability at most $\delta$. Then (see [4], Proposition 2.6):

$$\max_{I \in \mathcal{I}} \mathbb{E}_{\boldsymbol{q}}[C_\delta(I, A)] \geq \frac{1}{2} \min_{A \in \mathcal{A}} \mathbb{E}_{\boldsymbol{p}}[C_{2\delta}(I, A)] \tag{4}$$

## 2 Supplementary material for Section 5

### 2.1 Monochromatic Tessellation

We give a formal version of the claim about the monochromatic tessellation of Section 5:

**Theorem 7.** *Suppose we are given an ellipsoid $E$ such that $\frac{1}{d\Phi}E \subset \mathrm{conv}(S_C) \subset E$ for some stretch factor $\Phi > 0$. Then for a suitable choice of $\beta_i, \rho, b$, the tessellation $\mathcal{R}$ of the positive orthant of $E$ (Definition 3) satisfies:*

*(1) $|\mathcal{R}| \leq \max\left\{1, O\big(\frac{d\Phi}{\gamma} \ln\frac{d\Phi}{\gamma}\big)^d\right\}$*

*(2) $E \cap \mathbb{R}_+^d \subseteq \cup_{R \in \mathcal{R}} R$*

*(3) for every $R \in \mathcal{R}$, the set $R \cap E$ is monochromatic*

In order to prove Theorem 7, we define the tessellation and prove properties (1-3) for $\gamma \leq 1/2$. For $\gamma > \frac{1}{2}$ the tessellation is defined as for $\gamma = \frac{1}{2}$, and one can check all properties still hold. In the proof we use a constant $c = \sqrt{5}$ and assume $\gamma < c^2 - 2c$, which is satisfied since $c^2 - 2c = 5 - 2\sqrt{5} > 1/2$.

First of all, we define the intervals $T_i$. The base $i$-th coordinate is:

$$\beta_i = \frac{\gamma}{c\sqrt{2d}} \frac{L_i}{\Phi d} \tag{5}$$

Note that, for all $i$,

$$\frac{L_i}{\beta_i} = \frac{\Phi c d \sqrt{2d}}{\gamma} \tag{6}$$

Define:

$$\alpha = \frac{\gamma}{c\sqrt{2}\Phi d} \tag{7}$$

and let:

$$b = \max\left(0, \left\lceil \log_{1+\alpha}\left(\frac{c\Phi d\sqrt{2d}}{\gamma}\right) \right\rceil\right) \tag{8}$$

(The parameter $\rho$ of the informal description of Section 5 is exactly $1 + \alpha$). Finally, define the interval set along the $i$-th axis as:

$$
T_i = \begin{cases} \big\{ [0, \beta_i] \big\} & \text{if } b = 0 \\[2mm] \big\{ [0, \beta_i], (\beta_i, \beta_i(1+\alpha)], \ldots, (\beta_i(1+\alpha)^{b-1}, \beta_i(1+\alpha)^b] \big\} & \text{if } b \geq 1 \end{cases} \tag{9}
$$

**Proof of (1).** By construction, $|T_i| = b + 1$. Thus, $|\mathcal{R}| = \prod_{i \in [d]} |T_i| = (b+1)^d$. Thus, if $b = 0$ then $|\mathcal{R}| = 1$, else by (8) and 6,

$$
b = \left\lceil \frac{\ln\left(\frac{c\Phi d\sqrt{2d}}{\gamma}\right)}{\ln(1+\alpha)} \right\rceil \tag{10}
$$

$$
\leq \left\lceil \frac{2}{\alpha} \ln\left(\frac{c\Phi d\sqrt{2d}}{\gamma}\right) \right\rceil \qquad \text{since } \ln(1+\alpha) \geq {}^{\alpha}\!/_2 \text{ as } \alpha \leq 1 \tag{11}
$$

$$
= \left\lceil \frac{2\sqrt{2}c\Phi d}{\gamma} \ln \frac{c\Phi d\sqrt{2d}}{\gamma} \right\rceil \qquad \text{definition of } \alpha \tag{12}
$$

$$
= O\left(\frac{d\Phi}{\gamma} \ln \frac{d\Phi}{\gamma}\right) \qquad \text{since } d\Phi \geq 1, \gamma \leq {}^1\!/_2 \tag{13}
$$

in which case $|\mathcal{R}| = O\left(\frac{d\Phi}{\gamma} \ln \frac{d\Phi}{\gamma}\right)^d$. Taking the maximum over the two cases proves the claim.

**Proof of (2).** We show for any $x \in E \cap \mathbb{R}_+^d$ there exists $R \in \mathcal{R}$ containing $x$. Clearly, if $x \in E \cap \mathbb{R}_+^d$, then $\langle x, u_i \rangle \in [0, L_i]$ for all $i \in [d]$. But $T_i$ covers, along the $i$-th direction $u_i$, the interval from 0 to

$$
\beta_i(1+\alpha)^b = \beta_i(1+\alpha)^{\max(0, \lceil \log_{1+\alpha}(L_i/\beta_i) \rceil)} \geq \beta_i(1+\alpha)^{\lceil \log_{1+\alpha}(L_i/\beta_i) \rceil} \geq L_i \tag{14}
$$

Therefore some $R \in \mathcal{R}$ contains $x$.

**Proof of (3).** Given any hyperrectangle $R \in \mathcal{R}$, we show that the existence of $x, y \in R \cap E$ with $x \in C$ and $y \notin C$ leads to a contradiction. For the sake of the analysis we conventionally set the origin at the center $\mu$ of $E$, i.e. we assume $\mu = \mathbf{0}$.

We define $E_{\text{in}} = \frac{1}{\Phi d} E$ and let $M = U\Lambda U^\top$ be its PSD matrix, where $U = [u_1, \ldots, u_d]$ and $\Lambda = \text{diag}(\lambda_1, \ldots, \lambda_d)$. Note that $\lambda_i = \frac{1}{\ell_i^2} = \frac{\Phi^2 d^2}{L_i^2}$ where $\ell_i = \frac{L_i}{\Phi d}$ is the length of the $i$-th semiaxis of $E_{\text{in}}$. For any $R \in \mathcal{R}$, let $R_i$ be the projection of $R$ on $u_i$ (i.e. $R_i$ is one of the intervals of $T_i$ defined in (9)). Let $D = D(R) = \{i \in [d] : 0 \notin R_i\}$. We let $U_D$ and $U_{\neg D}$ be the matrices obtained by zeroing out the columns of $U$ corresponding to the indices in $[d] \setminus D$ and $D$, respectively. Observe that if $x, y \in R \cap E$ then:

$$
\langle x - y, u_i \rangle^2 < \alpha^2 \langle x, u_i \rangle^2 \quad \forall i \in D \tag{15}
$$

$$
\langle x - y, u_i \rangle^2 \leq \beta_i^2 \qquad \forall i \notin D \tag{16}
$$

Now suppose $C$ has margin at least $\gamma$ for some $\gamma \in (0, c^2 - 2c]$, and suppose $x, y \in R \cap E$ with $x \in C$ and $y \notin C$. Through a set of ancillary lemmata proven below, this leads to the absurd:

$$
\frac{\gamma^2}{c^2} < d_W(y, x)^2 \qquad\qquad\qquad \text{Lemma 5} \tag{17}
$$

$$
\leq d_M(y, x)^2 \qquad\qquad\qquad \text{Lemma 6} \tag{18}
$$

$$
< \alpha^2 d_M(x, \mu)^2 + \frac{\gamma^2}{2c^2} \qquad\qquad \text{Lemma 7} \tag{19}
$$

$$
\leq \frac{\gamma^2}{2c^2} + \frac{\gamma^2}{2c^2} \qquad\qquad\qquad \text{Lemma 8} \tag{20}
$$

In the rest of the proof we prove the four lemmata.

**Lemma 5.** $\frac{\gamma}{c} < d_W(y, x)$.

*Proof.* Let $\boldsymbol{z}$ be the point w.r.t. which the margin of $C$ holds. By the margin assumption,

$$d_W(\boldsymbol{y}, \boldsymbol{z}) > \sqrt{1 + \gamma} \qquad \text{and} \qquad d_W(\boldsymbol{x}, \boldsymbol{z}) \le 1 \tag{21}$$

By the triangle inequality then,

$$d_W(\boldsymbol{y}, \boldsymbol{x}) \ge d_W(\boldsymbol{y}, \boldsymbol{z}) - d_W(\boldsymbol{x}, \boldsymbol{z}) > \sqrt{1 + \gamma} - 1 \tag{22}$$

One can check that for $\gamma \le c^2 - 2c$ we have $1 + \gamma \ge (1 + \frac{\gamma}{c})^2$. Therefore

$$d_W(\boldsymbol{y}, \boldsymbol{x}) > \sqrt{(1 + \gamma/c)^2} - 1 = \frac{\gamma}{c} \tag{23}$$

as desired. $\qquad\square$

**Lemma 6.** $d_W(\cdot) \le d_M(\cdot)$.

*Proof.* By the assumptions of the theorem, $E_{\text{in}} \subseteq \text{conv}_{\boldsymbol{\mu}}(C)$. Moreover, by the assumptions on $d_W(\cdot)$, the unit ball of $d_W(\cdot)$ contains $\text{conv}(C)$. Thus, the unit ball of $d_W(\cdot)$ contains the unit ball of $d_M(\cdot)$. This implies $W \preceq M$, thus $\|\cdot\|_W \le \|\cdot\|_M$ and $d_W(\cdot) \le d_M(\cdot)$. $\qquad\square$

**Lemma 7.** $d_M(\boldsymbol{y}, \boldsymbol{x})^2 < \alpha^2 d_M(\boldsymbol{x}, \boldsymbol{\mu})^2 + \frac{\gamma^2}{2c^2}$.

*Proof.* We decompose $d_M(\boldsymbol{y}, \boldsymbol{x})^2$ along the colspaces of $U_D$ and $U_{\neg D}$:

$$d_M(\boldsymbol{y}, \boldsymbol{x})^2 = \|M^{1/2}(\boldsymbol{y} - \boldsymbol{x})\|_2^2 \tag{24}$$

$$= \|M^{1/2}(\boldsymbol{y} - \boldsymbol{x})\|_{U_D U_D^\top}^2 + \|M^{1/2}(\boldsymbol{y} - \boldsymbol{x})\|_{U_{\neg D} U_{\neg D}^\top}^2 \tag{25}$$

Next, we bound the two terms of (25). To this end, we need to show that for all $D \subseteq [d]$ and $\boldsymbol{v} \in \mathbb{R}^d$:

$$\|M^{1/2}\boldsymbol{v}\|_{U_D U_D^\top}^2 = \sum_{i \in D} \lambda_i \langle \boldsymbol{v}, \boldsymbol{u}_i \rangle^2 \tag{26}$$

Let indeed $J_D = \text{diag}(\mathbf{1}_D)$ be the selection matrix corresponding to the indices of $D$. Then $U_D = U J_D$, and so $U^\top U_D = U^\top U J_D = J_D$. This gives:

$$\|M^{1/2}\boldsymbol{v}\|_{U_D U_D^\top}^2 = \boldsymbol{v}^\top (U \Lambda^{1/2} U^\top) U_D U_D^\top (U \Lambda^{1/2} U^\top) \boldsymbol{v} \quad \text{definition of } M \text{ and } \|\cdot\|. \tag{27}$$

$$= \boldsymbol{v}^\top U \Lambda^{1/2} J_D J_D \Lambda^{1/2} U^\top \boldsymbol{v} \qquad\qquad \text{since } U^\top U_D = J_D \tag{28}$$

$$= \boldsymbol{v}^\top U J_D \Lambda^{1/2} \Lambda^{1/2} J_D U^\top \boldsymbol{v} \qquad\qquad \text{since } \Lambda, J_D \text{ are diagonal} \tag{29}$$

$$= \boldsymbol{v}^\top U_D \Lambda U_D^\top \boldsymbol{v} \qquad\qquad\qquad \text{since } U J_D = U_D \tag{30}$$

$$= \|U_D^\top \boldsymbol{v}\|_\Lambda^2 \qquad\qquad\qquad\qquad \text{by definition} \tag{31}$$

$$= \sum_{i \in D} \lambda_i \langle \boldsymbol{v}, \boldsymbol{u}_i \rangle^2 \tag{32}$$

Now we can bound the first term of (25):

$$\|M^{1/2}(\boldsymbol{y} - \boldsymbol{x})\|_{U_D U_D^\top}^2 = \sum_{i \in D} \lambda_i \langle \boldsymbol{y} - \boldsymbol{x}, \boldsymbol{u}_i \rangle^2 \qquad\qquad \text{by (32)} \tag{33}$$

$$< \alpha^2 \sum_{i \in D} \lambda_i \langle \boldsymbol{x}, \boldsymbol{u}_i \rangle^2 \qquad\qquad\qquad \text{by (15)} \tag{34}$$

$$= \alpha^2 \|M^{1/2}\boldsymbol{x}\|_{U_D U_D^\top}^2 \qquad\qquad\qquad \text{by (32)} \tag{35}$$

$$\le \alpha^2 \|M^{1/2}\boldsymbol{x}\|_{U U^\top}^2 \tag{36}$$

$$= \alpha^2 \|M^{1/2}\boldsymbol{x}\|_2^2 \qquad\qquad\qquad\qquad \text{since } U U^\top = I \tag{37}$$

$$= \alpha^2 d_M^2(\boldsymbol{x}, \boldsymbol{\mu}) \qquad\qquad\qquad\qquad \text{since } \boldsymbol{\mu} = \mathbf{0} \tag{38}$$

And for the second term of (25), we have:

$$\|M^{1/2}(\boldsymbol{y} - \boldsymbol{x})\|^2_{U_{\neg D} U^\top_{\neg D}} = \sum_{i \notin D} \lambda_i \langle \boldsymbol{y} - \boldsymbol{x}, \boldsymbol{u}_i \rangle^2 \qquad \text{by (32)} \tag{39}$$

$$\leq \sum_{i \notin D} \lambda_i \beta_i^2 \qquad \text{by (16)} \tag{40}$$

$$= \sum_{i \notin D} \frac{\Phi^2 d^2}{L_i^2} \left( \frac{\gamma}{c\sqrt{2d}} \frac{L_i}{\Phi d} \right)^2 \qquad \text{by definition of } \lambda_i \text{ and } \beta_i \tag{41}$$

$$= \sum_{i \notin D} \frac{\gamma^2}{2dc^2} \tag{42}$$

$$\leq \frac{\gamma^2}{2c^2} \tag{43}$$

Summing the bounds on the two terms shows that $d_M(\boldsymbol{y}, \boldsymbol{x})^2 < \alpha^2 d_M(\boldsymbol{x}, \boldsymbol{\mu})^2 + \frac{\gamma^2}{2c^2}$, as claimed. $\qquad\square$

**Lemma 8.** $\alpha^2 d_M(\boldsymbol{x}, \boldsymbol{\mu})^2 \leq \frac{\gamma^2}{2c^2}$.

*Proof.* By construction we have $\boldsymbol{x} \in E$ and $E = \Phi d \cdot E_{\text{in}}$. Therefore $\frac{1}{\Phi d}\boldsymbol{x} \in E_{\text{in}}$, that is:

$$1 \geq d_M \left( \frac{1}{\Phi d}\boldsymbol{x}, \boldsymbol{\mu} \right)^2 = \frac{1}{\Phi^2 d^2} d_M(\boldsymbol{x}, \boldsymbol{\mu})^2 \tag{44}$$

where we used the fact that $d_M(\cdot, \boldsymbol{\mu})^2 = \| \cdot \|^2_M$ since $\boldsymbol{\mu} = \boldsymbol{0}$. Rearranging terms, this proves that $d_M(\boldsymbol{x}, \boldsymbol{\mu})^2 \leq \Phi^2 d^2$. Multiplying by $\alpha^2$, we obtain:

$$\alpha^2 d_M(\boldsymbol{x}, \boldsymbol{\mu})^2 \leq \left( \frac{\gamma}{\sqrt{2}c\Phi d} \right)^2 \Phi^2 d^2 = \frac{\gamma^2}{2c^2} \tag{45}$$

as desired. $\qquad\square$

The proof of the theorem is complete.

## 2.2 Low-stretch separators and proof of Theorem 3

In this section we show how to compute the separator of Theorem 3. In fact, computing the separator is easy; the nontrivial part is Theorem 3 itself, that is, showing that such a separator always exists.

To compute the separator we first compute the MVEE $E_J = (M^\star, \boldsymbol{\mu}^\star)$ of $S_C$ (see Section 5). We then solve the following semidefinite program:

$$\begin{aligned} \max_{\alpha \in \mathbb{R}, \boldsymbol{\mu} \in \mathbb{R}^d, M \in \mathbb{R}^{d \times d}} \quad & \alpha \\ \text{s.t.} \quad & M \succeq \alpha M^\star \\ & \langle M, (\boldsymbol{x} - \boldsymbol{\mu})(\boldsymbol{x} - \boldsymbol{\mu})^\top \rangle \leq 1 \quad \forall \boldsymbol{x} \in S_C \\ & \langle M, (\boldsymbol{y} - \boldsymbol{\mu})(\boldsymbol{y} - \boldsymbol{\mu})^\top \rangle > 1 \quad \forall \boldsymbol{y} \in S_{\overline{C}} \end{aligned} \tag{46}$$

where, for any two symmetric matrices $A$ and $B$, $\langle A, B \rangle = \text{tr}(AB)$ is the usual Frobenius inner product, implying $\langle M, (\boldsymbol{x} - \boldsymbol{\mu})(\boldsymbol{x} - \boldsymbol{\mu})^\top \rangle = d_M(\boldsymbol{x}, \boldsymbol{\mu})^2$. In words, the constraint $M \succeq \alpha M^\star$ says that $E$ must fit into $E_J$ if we scale $E_J$ by a factor $\Phi = 1/\sqrt{\alpha}$. The other constraints require $E$ to contain all of $S_C$ but none of the points of $S_{\overline{C}}$. The objective function thus minimizes the stretch $\Phi$ of $E$.

In the rest of this paragraph we prove Theorem 3.

**Proof of Theorem 3 (sketch).** To build the intuition, we first give a proof sketch where the involved quantities are simplified. The analysis is performed in the latent space $\mathbb{R}^d$ with inner product $\langle \boldsymbol{u}, \boldsymbol{v} \rangle = \boldsymbol{u}^\top W \boldsymbol{v}$. Setting conventionally $\boldsymbol{z} = \boldsymbol{0}$, $C$ then lies in the unit ball $\mathcal{B}_0$ and all points of $X \setminus C$ lie outside $\sqrt{1+\gamma}\,\mathcal{B}_0$. For simplicity we assume $\gamma \ll 1$ so that $\sqrt{1+\gamma} \simeq 1 + \gamma$, but we can easily extend the result to any $\gamma > 0$. Now fix the subset $S_C \subseteq C$, and let $E_J = E_J(S_C)$ be the MVEE of $S_C$. Observe the following fact: $\mathcal{B}_0$ trivially satisfies (1), but in general violates (2); in contrast, $E_J$ trivially satisfies (2), but in general violates (1). The key idea is thus to "compare" $\mathcal{B}_0$ and $E_J$ and take, loosely speaking, the best of the two. To see how this works, suppose for instance $E_J$ has small radius, say less than $\gamma/4$. In this case, $E = E_J$ yields the thesis. Indeed, since the center $\boldsymbol{\mu}^\star$ of $E_J$ is in $\mathcal{B}_0$, then any point of $E$ is within distance $1 + \gamma/4 \leq \sqrt{1+\gamma}$ of the center of $\mathcal{B}_0$, and lies inside $\sqrt{1+\gamma}\,\mathcal{B}_0$. Thus $E_J$ separates $S_C$ from $X \setminus C$, satisfying (1). At the other extreme, suppose $E_J$ is large, say with all its $d$ semiaxes longer than $\gamma/4$. In this case, $E = \mathcal{B}_0$ yields the thesis: indeed, by hypothesis $E$ fits entirely inside $4/\gamma\, E_J$, satisfying (2). Unfortunately, the general case is more complex, since $E_J$ may be large along some axes and small along others. In this case, both $\mathcal{B}_0$ and $E_J$ fail to satisfy the properties. This requires us to choose the axes and the center of $E$ more carefully. We show how to do this with the help of Figure 1.

Figure 1: Left: the MVEE $E_J$ of $S_C$ and the affine subspace $U + \boldsymbol{\mu}^\star$ (marked simply as $U$) spanned by its largest semiaxes. There is no guarantee that $E_J \subseteq \sqrt{1+\gamma}\,\mathcal{B}_0$. Right: the separator $E$, centered in the center $\boldsymbol{\mu}$ of $B$, with the largest semiaxis in $U$ and the smallest one in $U_\perp$. We can guarantee that $S_C \subset E \subset \sqrt{1+\gamma}\,\mathcal{B}_0$.

Let $\{\boldsymbol{u}_1, \ldots, \boldsymbol{u}_d\}$ be the orthonormal basis defined by the semiaxes of $E_J$ and $\ell_1^\star, \ldots, \ell_d^\star$ be the corresponding semiaxes lengths. We define a threshold $\varepsilon = \gamma^3/d^2$, and partition $\{\boldsymbol{u}_1, \ldots, \boldsymbol{u}_d\}$ as $A_P = \{i \,:\, \ell_i^\star > \varepsilon\}$ and $A_Q = \{i \,:\, \ell_i^\star \leq \varepsilon\}$. Thus $A_P$ contains the large semiaxes of $E_J$ and $A_Q$ the small ones. Let $U, U_\perp$ be the subspaces spanned by $\{\boldsymbol{u}_i \,:\, i \in A_P\}$ and $\{\boldsymbol{u}_i \,:\, i \in A_Q\}$, respectively. Consider the subset $B = \mathcal{B}_0 \cap (\boldsymbol{\mu}^\star + U)$. Note that $B$ is a ball in at most $d$ dimensions, since it is the intersection of a $d$-dimensional ball and an affine linear subspace of $\mathbb{R}^d$. Let $\boldsymbol{\mu}$ and $\ell$ be, respectively, the center and radius of $B$. We set the center of $E$ at $\boldsymbol{\mu}$, and the lengths $\ell_i$ of its semiaxes as follows:

$$\ell_i = \begin{cases} \dfrac{\ell}{\sqrt{1-\gamma}} & \text{if } i \in A_P \\[2mm] \dfrac{\ell_i^\star}{\sqrt{\varepsilon}} & \text{if } i \in A_Q \end{cases} \tag{47}$$

Loosely speaking, we are "copying" the semiaxes from either $\mathcal{B}_0$ or $E_J$ depending on $\ell_i^\star$. In particular, the large semiaxes (in $A_P$) are set so to contain all of $B$ and exceed it by a little, taking care of not intersecting $\sqrt{1+\gamma}\,\mathcal{B}_0$. Instead, the small semiaxes (in $A_Q$) are so small that we can safely set them to $1/\sqrt{\varepsilon}$ times those of $E_J$, so that we add some "slack" to include $S_C$ without risking to intersect $\sqrt{1+\gamma}\,\mathcal{B}_0$. Now we are done, and our low-stretch separator is $(M, \boldsymbol{\mu})$ where $M = \sum_{i=1}^d \ell_i^{-2} \boldsymbol{u}_i \boldsymbol{u}_i^\mathsf{T}$. This the ellipsoid $E$ that yields Theorem 3. In the next paragraph, we show how we can find efficiently all points in $E$ that belong to $C$.

## 2.3 Proof of Theorem 3 (full).

We prove the theorem for $\gamma \leq 1/5$ and use the fact that whenever $C$ has weak margin $\gamma$ then it also has weak margin $\gamma'$ for all $\gamma' > \gamma$. As announced, the analysis is carried out in the latent space $\mathbb{R}^d$ equipped with the inner product $\langle \boldsymbol{u}, \boldsymbol{v} \rangle = \boldsymbol{u}^\top W \boldsymbol{v}$. All norms $\|\boldsymbol{u}\|$, distances $d(\boldsymbol{u}, \boldsymbol{v})$, and (cosine of) angles $\langle \boldsymbol{u}, \boldsymbol{v} \rangle / (\|\boldsymbol{u}\| \, \|\boldsymbol{v}\|)$ are computed according to this inner product unless otherwise specified. Let $\mathcal{B}_0$ be the unit ball centered at the origin, which we conventionally set at $\boldsymbol{z}$, the point in the convex hull of $C$ according to which the margin is computed. Then, by assumption, $C \subset \mathcal{B}_0$, and $\boldsymbol{x} \notin \sqrt{1 + \gamma} \, \mathcal{B}_0$ for all $\boldsymbol{x} \notin C$. For ease of notation, in this proof be denote the MVEE by $E^\star$ rather than $E_{\mathrm{J}}$. Let then $(E^\star, \boldsymbol{\mu}^\star)$ be the MVEE of $S_C$; note that $\boldsymbol{\mu}^\star \in \mathrm{conv}(S_C) \subseteq \mathcal{B}_0$. We let $\boldsymbol{u}_1, \dots, \boldsymbol{u}_d$ be the orthonormal eigenvector basis given by the axes of $E^\star$ and $\lambda_1^\star, \dots, \lambda_d^\star$ the corresponding eigenvalues. Note that if $\min_i \lambda_i^\star \geq 5/\gamma^2$ then $E^\star$ has radius $\leq \gamma/\sqrt{5}$ and thus, since $\boldsymbol{\mu}^\star \in \mathcal{B}_0$ and $\gamma \leq 1/5$, its distance from $\mathcal{B}_0$ is at most $1 + \gamma/\sqrt{5} = \sqrt{1 + 2\gamma/\sqrt{5} + \gamma^2/5} < \sqrt{1 + \gamma}$. In this case we can simply set $E = E^\star$ and the thesis is proven. Thus, from now on we assume $\min_i \lambda_i^\star < 5/\gamma^2$.

Figure 2: Left: the separating ball $\mathcal{B}_0$ of $C$, the MVEE $E^\star$ of $S_C$, and the affine subspace $U + \boldsymbol{\mu}^\star$ spanned by its largest semiaxes. Middle: $E$ is our separator centered in the center $\boldsymbol{\mu}$ of the ball $B = U \cap \mathcal{B}_0$. Right: a point $\boldsymbol{x} \in S_C$ with its projections onto $U$ and $U_\perp$ with respect to the origin, which we conventionally set at $\boldsymbol{\mu}$ (the center of $E$).

Now let:

$$\varepsilon = \frac{\gamma^3}{32d^2} \tag{48}$$

and partition (the indices of) the basis $\{\boldsymbol{u}_1, \dots, \boldsymbol{u}_d\}$ as follows:

$$A_P = \{i \,:\, \lambda_i^\star < 1/\varepsilon^2\}, \quad A_Q = [d] \setminus A_P \tag{49}$$

Since $\min_i \lambda_i^\star < 5/\gamma^2$ and $5/\gamma^2 \leq 1/\varepsilon^2$, then by construction the set $A_P$ is not empty. We now define the ellipsoid $E$. Let $U, U_\perp$ be the subspaces spanned by $\{\boldsymbol{u}_i : i \in A_P\}$ and $\{\boldsymbol{u}_i : i \in A_Q\}$ respectively, and let $B = \mathcal{B}_0 \cap (\boldsymbol{\mu}^\star + U)$. Note that $B$ is a ball, since it is the intersection of a ball and an affine linear subspace. Let $\boldsymbol{\mu}$ and $\ell$ be, respectively, the center and radius of $B$ and define

$$\lambda_i = \begin{cases} (1 - \sqrt{5\gamma/4})\ell^{-2} & i \in A_P \\ \varepsilon \lambda_i^\star & i \in A_Q \end{cases} \qquad M = \sum_{i=1}^d \lambda_i \boldsymbol{u}_i \boldsymbol{u}_i^\mathsf{T} \tag{50}$$

Then our ellipsoidal separator is $E = \{\boldsymbol{x} \in \mathbb{R}^d \,:\, d_M(\boldsymbol{x}, \boldsymbol{\mu}) \leq 1\}$. See Figure 2 for a pictorial representation. We now prove that $E$ satisfies: **(1)** $S_C \subset E$, **(2)** $E \subseteq \frac{64\sqrt{2}d^2}{\gamma^3} E^\star(S_C)$, **(3)** $E \subset \sqrt{1 + \gamma} \, \mathcal{B}_0$.

**Proof of (1).** Set the center $\boldsymbol{\mu}$ of $E$ as the origin. For all $i \in [d]$ let $U_i = \boldsymbol{u}_i \boldsymbol{u}_i^{\mathsf{T}}$ and define the following matrices:

$$P_0 = \sum_{i \in A_P} U_i, \qquad Q_0 = \sum_{i \in A_Q} U_i \tag{51}$$

$$P = \sum_{i \in A_P} \lambda_i U_i, \qquad Q = \sum_{i \in A_Q} \lambda_i U_i \tag{52}$$

$$P_\star = \sum_{i \in A_P} \lambda_i^\star U_i, \qquad Q_\star = \sum_{i \in A_Q} \lambda_i^\star U_i \tag{53}$$

We want to show that $d_M^2(\boldsymbol{x}, \boldsymbol{\mu}) \le 1$ for all $\boldsymbol{x} \in S_C$. Note that $d_M(\boldsymbol{x}, \boldsymbol{\mu})^2$ equals (recall that $\boldsymbol{\mu} = \boldsymbol{0}$):

$$\boldsymbol{x}^{\mathsf{T}} P \boldsymbol{x} + \boldsymbol{x}^{\mathsf{T}} Q \boldsymbol{x} \tag{54}$$

Let us start with the second term of (54). By definition of $Q_\star$ and since $\boldsymbol{\mu}^{\star \mathsf{T}} Q_\star = (\boldsymbol{\mu}^\star - \boldsymbol{\mu})^{\mathsf{T}} Q_\star = \boldsymbol{0}$ because $\boldsymbol{\mu}^\star - \boldsymbol{\mu} \in U$,

$$\boldsymbol{x}^{\mathsf{T}} Q \boldsymbol{x} = \varepsilon \, \boldsymbol{x}^{\mathsf{T}} Q_\star \boldsymbol{x} = \varepsilon \, (\boldsymbol{x} - \boldsymbol{\mu}^\star)^{\mathsf{T}} Q_\star (\boldsymbol{x} - \boldsymbol{\mu}^\star) \le \varepsilon < \frac{\gamma}{4} \tag{55}$$

where the penultimate inequality follows from $\boldsymbol{x} \in E^\star$.

We turn to the first term of (54). If we let $\boldsymbol{p}, \boldsymbol{q}$ be the projections of $\boldsymbol{x} - \boldsymbol{\mu} = \boldsymbol{x}$ onto $U, U_\perp$, so that

$$\|\boldsymbol{p}\|^2 = \boldsymbol{x}^{\mathsf{T}} P_0 \boldsymbol{x}, \qquad \|\boldsymbol{q}\|^2 = \boldsymbol{x}^{\mathsf{T}} Q_0 \boldsymbol{x} \tag{56}$$

then by definition of the $\lambda_i$ we have:

$$\boldsymbol{x}^{\mathsf{T}} P \boldsymbol{x} = \frac{1 - \sqrt{5\gamma/4}}{\ell^2} \|\boldsymbol{p}\|^2 \tag{57}$$

We can thus focus on bounding $\|\boldsymbol{p}\|$. Since $B$ is a ball of radius $\ell$, then $\|\boldsymbol{p}\| \le \ell + d(\boldsymbol{p}, B)$, where $d(\boldsymbol{p}, B)$ is the distance of $\boldsymbol{p}$ from its projection on $B$ —see Figure 3, left.

Figure 3: Left: a point $\boldsymbol{x} \in S_C \subset \mathcal{B}_0$ which lies in $E$ as well. Right: for a fixed $a > 0$, the ratio $b/a$ is maximized when the segment of length $a$ lies on the line passing through the center of $\mathcal{B}_0$, in which case $b/a = \frac{\sin \theta}{1 - \cos \theta}$ for some $\theta \in (0, \pi/2)$.

Now, since $\boldsymbol{x} \in \mathcal{B}_0$, the ratio $\frac{d(\boldsymbol{p}, B)}{\|\boldsymbol{q}\|}$ is maximized when $\ell \to 0$ (i.e., $B$ has a vanishing radius), in which case $d(\boldsymbol{p}, B) \le \sin \theta$ and $\|\boldsymbol{q}\| \ge 1 - \cos \theta$, where $\theta \in (0, \pi/2]$; see Figure 3 right. Then:

$$\frac{\|\boldsymbol{q}\|}{d(\boldsymbol{p}, B)} \ge \frac{1 - \cos \theta}{\sin \theta} = \tan \frac{\theta}{2} \ge \frac{\theta}{2} \ge \frac{\sin \theta}{2} \ge \frac{d(\boldsymbol{p}, B)}{2} \tag{58}$$

where we used the tangent half-angle formula and the Taylor expansion of $\tan \theta$. This yields $d(\boldsymbol{p}, B) \le \sqrt{2 \|\boldsymbol{q}\|_2}$. Thus:

$$\|\boldsymbol{p}\| \le \ell + \sqrt{2\|\boldsymbol{q}\|} \tag{59}$$

But since $\lambda_i^\star \geq 1/\varepsilon^2$ for all $i \in A_Q$:

$$\|\boldsymbol{q}\|^2 = \boldsymbol{x}^\mathsf{T} Q_0 \boldsymbol{x} \leq \varepsilon^2 \, \boldsymbol{x}^\mathsf{T} Q_\star \boldsymbol{x} = \varepsilon^2 (\boldsymbol{x} - \boldsymbol{\mu}^\star)^\mathsf{T} Q_\star (\boldsymbol{x} - \boldsymbol{\mu}^\star) \leq \varepsilon^2 \tag{60}$$

Therefore:

$$\boldsymbol{x}^\mathsf{T} P \boldsymbol{x} \leq \frac{1 - \sqrt{5\gamma/4}}{\ell^2}\left(\ell + \sqrt{2\varepsilon}\right)^2 \leq \left(1 - \sqrt{5\gamma/4}\right)\left(1 + \sqrt{2\varepsilon}/\ell\right)^2 \tag{61}$$

Next, we show that $\frac{\sqrt{2\varepsilon}}{\ell} \leq \frac{1}{2}\sqrt{5\gamma/4}$. First,

$$\sqrt{2\varepsilon} = \sqrt{2\frac{\gamma^3}{32 d^2}} = \frac{\gamma\sqrt{\gamma}}{4d} \tag{62}$$

We now temporarily set $\boldsymbol{\mu}^\star$ as the origin. We want to show that the projection of $1/d\, E^\star$ on $U$ is contained in $B$. Now, the projection of an ellipsoid on the subspace spanned by a subset of its axes is a subset of the ellipsoid itself, and $U$ is by definition spanned by a subset of the axes of $E^\star$. Therefore the projection $P$ of $1/d\, E^\star$ on $U$ satisfies $P \subseteq 1/d\, E^\star$. Suppose then by contradiction that $P \not\subseteq B$. Since $B = U \cap \mathcal{B}_0$, this implies that $1/d\, E^\star \not\subseteq \mathcal{B}_0$. But by John's theorem, $1/d\, E^\star \subseteq \mathrm{conv}(S_C)$, and therefore $\mathrm{conv}(S_C) \not\subseteq \mathcal{B}_0$, which is absurd. Therefore $P \subseteq B$.

Let us get back to the proof, with $\boldsymbol{\mu}$ as the origin. On the one hand, the definitions of $A_P$ and $U$ imply that the largest semiaxis of $E^\star$ of length $\ell^\star = 1/\sqrt{\min_i \lambda_i^\star}$ lies in $U$, thus $P$ has radius at least $\frac{1}{d}\ell^\star$. On the other hand $B$ has radius $\ell$, and we have seen that $P \subseteq B$. Therefore, $\ell \geq \frac{1}{d}\ell^\star$. Finally, by our assumption on $\min_i \lambda_i^\star$, we have $\min_i \lambda_i^\star < 5/\gamma^2$ and so $\ell^\star > \gamma/\sqrt{5}$. Therefore, $\ell \geq \gamma/\sqrt{5}d$, which together with (62) guarantees $\frac{\sqrt{2\varepsilon}}{\ell} \leq \frac{\sqrt{5\gamma}}{4} = \frac{1}{2}\sqrt{5\gamma/4}$. Thus, continuing (61):

$$\boldsymbol{x}^\mathsf{T} P \boldsymbol{x} \leq (1 - \sqrt{5\gamma/4})\left(1 + \frac{1}{2}\sqrt{5\gamma/4}\right)^2 \tag{63}$$

Now $(1-x)(1+\frac{x}{2})^2 < 1 - \frac{3}{4}x^2$ for all $x > 0$, thus with $x = \sqrt{5\gamma/4} > \sqrt{\gamma}$ we get:

$$\boldsymbol{x}^\mathsf{T} P \boldsymbol{x} < 1 - \frac{3}{4}\gamma \tag{64}$$

By summing (55) and (64), we get:

$$\boldsymbol{x}^\mathsf{T} P \boldsymbol{x} + \boldsymbol{x}^\mathsf{T} Q \boldsymbol{x} < 1 - \frac{3}{4}\gamma + \frac{\gamma}{4} < 1 \tag{65}$$

**Proof of (2).** Comparing the eigenvalues of $E$ and $E^\star$, and using $\ell \leq 1$ and $\gamma \leq 1/5$, we obtain:

$$\frac{\lambda_i}{\lambda_i^\star} \geq \begin{cases} \frac{(1 - \sqrt{5\gamma/4})/\ell^2}{1/\varepsilon^2} \geq \frac{\varepsilon^2}{2} & i \in A_P \\ \varepsilon > \frac{\varepsilon^2}{2} & i \in A_Q \end{cases} \tag{66}$$

Thus the semiaxes lengths of $E$ are at most $\sqrt{2}/\varepsilon$ times those of $E^\star$. Now let $E_+^\star$ be the set obtained by scaling $E^\star$ by a factor $2\sqrt{2}/\varepsilon = 64\sqrt{2}d^2/\gamma^3$ about its origin $\boldsymbol{\mu}^\star$. Note that $\boldsymbol{\mu}^\star \in \mathrm{conv}(S_C)$ and, by item **(1)**, $\mathrm{conv}(S_C) \subseteq E$, which implies $\boldsymbol{\mu}^\star \in E$. Now, $E_+^\star$ contains any set of the form $\boldsymbol{y} + \frac{1}{2}E_+^\star$ if the latter contains $\boldsymbol{\mu}^\star$; this includes the set $\frac{\sqrt{2}}{\varepsilon}E^\star$ centered in $\boldsymbol{\mu}$, which in turn contains $E$ as we already said.

**Proof of (3).** We prove that $d(\boldsymbol{x}, \mathcal{B}_0)^2 < \gamma$ for all $\boldsymbol{x} \in E$. Since $\mathcal{B}_0$ is the unit ball, this implies $E \subset \sqrt{1+\gamma}\,\mathcal{B}_0$. Consider then any such $\boldsymbol{x}$. Let again $\boldsymbol{p}, \boldsymbol{q}$ be the projections of $\boldsymbol{x}$ on $U$ and $U_\perp$ respectively. Because $B \subseteq \mathcal{B}_0$, $d(\boldsymbol{x}, \mathcal{B}_0)^2 \leq d(\boldsymbol{x}, B)^2 = d(\boldsymbol{p}, B)^2 + \|\boldsymbol{q}\|^2$. See again Figure 3, left, but with $\boldsymbol{x}$ possibly outside $\mathcal{B}_0$. For the first term, note that

$$d(\boldsymbol{p}, B) \leq \max_{i \in A_P} \sqrt{1/\lambda_i} - \ell \tag{67}$$

By definition of $\lambda_i$, this yields:

$$d(\boldsymbol{p}, B)^2 \leq \left(\frac{\ell}{\sqrt{1 - \sqrt{5\gamma/4}}} - \ell\right)^2 \leq \left(\frac{1}{\sqrt{1 - \sqrt{5\gamma/4}}} - 1\right)^2 \qquad \text{(because } \ell \leq 1\text{)}$$

Now we show that the right-hand side is bounded by $\frac{3}{4}\gamma$. Consider $f(x) = \frac{1}{\sqrt{1-x}} - 1$ for $x \in [0, 1/2]$. Now $\frac{\partial^2 f}{\partial x^2} = \frac{3}{4}(1-x)^{-5/2} > 0$, so $f$ is convex. Moreover, $f(1/2) = \sqrt{2} - 1 < 0.83 \cdot 1/2$, and clearly $f(0) = 0 \le 0.83 \cdot 0$. By convexity then, for all $x \in [0, 1/2]$ we have $f(x) \le 0.83\, x$ which implies $f(x)^2 < 0.75\, x^2$. By substituting $x = \sqrt{5\gamma/4}$, for all $\gamma \le 1/5$ we obtain:

$$d(\boldsymbol{p}, B)^2 \le \left( \frac{1}{\sqrt{1 - \sqrt{5\gamma/4}}} - 1 \right)^2 < \frac{3}{4} \cdot \frac{5}{4}\gamma = \frac{15}{16}\gamma \tag{68}$$

Let us now turn to $\boldsymbol{q}$. By definition of $Q_0$, of $Q$, and of $\lambda_i$ for $i \in A_Q$, we have:

$$\|\boldsymbol{q}\|^2 = \boldsymbol{x}^\mathsf{T} Q_0 \boldsymbol{x} \le \max_{i \in A_Q} \frac{1}{\lambda_i} \boldsymbol{x}^\mathsf{T} Q \boldsymbol{x} = \max_{i \in A_Q} \frac{1}{\varepsilon \lambda_i^\star} \boldsymbol{x}^\mathsf{T} Q \boldsymbol{x} \tag{69}$$

But $\boldsymbol{x}^\mathsf{T} Q \boldsymbol{x} \le 1$ since $\boldsymbol{x} \in E$, and recalling that $\lambda_i^\star \ge 1/\varepsilon^2$ for all $i \in A_Q$, we obtain:

$$\|\boldsymbol{q}\|^2 \le \frac{1}{\varepsilon(1/\varepsilon^2)} = \varepsilon < \frac{\gamma}{16} \tag{70}$$

Finally, by summing (68) and (70):

$$d(\boldsymbol{x}, \mathcal{B}_0)^2 \le d(\boldsymbol{p}, B)^2 + \|\boldsymbol{q}\|^2 < \gamma \tag{71}$$

The proof is complete.

# 3 Supplementary material for Section 6

## 3.1 Lemma 9

**Lemma 9.** *Let $b > 0$ be a sufficiently large constant. Let $S$ be a sample of points drawn independently and uniformly at random from $X$. Let $C = \arg\max_{C_j \in \mathcal{C}} |S \cap C_j|$, let $S_C = S \cap C$, and suppose $|S_C| \ge bd^2 \ln k$. If $E$ is any (possibly degenerate) ellipsoid in $\mathbb{R}^d$ such that $S_C = C \cap E$, then with probability at least $1/2$ we have $|C \cap E| \ge |X|\frac{1}{4k}$. The same holds if we require that $E \cap (S \setminus S_C) = \emptyset$, i.e., that $E$ separates $S_C$ from $S \setminus S_C$.*

*Proof.* Let $n = |X|$ for short, and for any ellipsoid $E$ let $E_X = E \cap X$. We show that, with $C$ defined as above, **(i)** with probability at least $1 - 1/4$ we have $|C| \ge n/2k$, and **(ii)** with probability at least $1 - 1/4$, if $|C| \ge n/2k$ then $|E_X \triangle C| \le 1/2|C|$ where $\triangle$ denotes symmetric difference. By a union bound, then, with probability at least $1/2$ we have $|E \cap C| \ge |C| - |E_X \triangle C| \ge \frac{1}{2}|C| \ge n/4k$.

**(i)**. Let $S$ be the multiset of samples drawn from $X$, and for every cluster $C_i \in \mathcal{C}$ let $N_i$ be the number of samples in $C_i$. Let $s = kbd^2 \ln k$; note that $|S| \le s$ since there are at most $k$ clusters. Now fix any $C_i$ with $|C_i| < \frac{n}{2k}$. Then $\mathbb{E}[N_i] \le s\frac{|C_i|}{n} < \frac{bd^2 \ln k}{2}$, and by standard concentration bounds (Lemma 4 in this supplementary material), we have $\mathbb{P}(N_i \ge bd^2 \ln k) = \exp(-\Omega(b \ln k))$, which for $b$ large enough drops below $1/4k$. Therefore, the probability that $N_i \ge bd^2 \ln k$ when taking $s \le kbd^2 \ln k$ samples is at most $1/4k$. By a union bound on all $C_i$ with $|C_i| < n/2k$, then, $|C| \ge n/2k$ with probability $1 - 1/4$.

**(ii)**. Consider now any $C_i$ with $|C_i| \ge n/2k$. We invoke the generalization bounds of Theorem 6 in this supplementary material with $\varepsilon = 1/4k$ and $\delta = 1/4k$, on the hypothesis class $\mathcal{H}$ of all (possibly degenerate) ellipsoids in $\mathbb{R}^d$. For $b$ large enough, the generalization error of any ellipsoid $E$ that contains $S_C$ is, with probability at least $1 - 1/4k$, at most $1/4k$, which means $|E_X \triangle C_i| \le n/4k \le 1/2|C_i|$, as desired. By a union bound on all clusters, with probability at least $1 - 1/4$ this holds for all $C_i$ with $|C_i| \ge n/2k$. The same argument holds if we require $E$ to separate $S \cap C_i$ from $S \setminus C_i$, see again Theorem 6. By a union bound with point (i) above, we have $E \cap C \le 1/2|C|$ with probability at least $1/2$, as claimed. □

## 3.2 Proof of Lemma 3

Let $X_0 = X$ and $N_0 = n$, and for all $i \ge 1$, let $X_i$ be the set of points not yet labeled at the end of round $i$, let $N_i = |X_i|$, and let $R_i = \mathbb{I}\{N_i \le N_{i-1}(1 - 1/4k)\}$. Recall that $S_C$ is large enough

so that, by Lemma 9 in this supplementary material, we have $\mathbb{P}(R_i = 1 \mid X_{i-1}) \geq 1/2$ for all $i$. For every $t \geq 1$ let $\rho_t = \sum_{i=1}^t R_i$. Note that:

$$N_t \leq N_0(1 - 1/4k)^{\rho_t} < ne^{-\frac{\rho_t}{4k}} \tag{72}$$

If $\rho_t \geq 4k\ln(1/\varepsilon)$, then $N_t < \varepsilon n$ and $\text{RECUR}(X, k, \gamma, \varepsilon)$ stops. The number of rounds executed by $\text{RECUR}(X, k, \gamma, \varepsilon)$ is thus at most $r_\varepsilon = \min\{t : \rho_t \geq 4k\ln(1/\varepsilon)\}$.

Now, for all $i \geq 1$ consider the $\sigma$-algebra $\mathcal{F}_{i-1}$ generated by $X_0, \ldots, X_{i-1}$, and define: $Z_i = R_i B_i$, where $B_1, B_2, \ldots$ are Bernoulli random variables where each $B_i$ has parameter $1/(2\,\mathbb{E}[R_i \mid \mathcal{F}_{i-1}])$. Obviously, $Z_i \leq R_i$, and thus for all $t$ we deterministically have:

$$\rho_t = \sum_{i=1}^t R_i \geq \sum_{i=1}^t Z_i \tag{73}$$

Now note that:

$$\mathbb{E}[Z_i \mid \mathcal{F}_{i-1}] = \mathbb{E}[R_i \mid \mathcal{F}_{i-1}]\frac{1}{2\,\mathbb{E}[R_i \mid \mathcal{F}_{i-1}]} = \frac{1}{2} \tag{74}$$

Now we can prove the theorem. For the first claim, simply note that $\mathbb{E}[r_\varepsilon] \leq 8k\ln(1/\varepsilon)$, as this is the expected number of fair coin tosses to get $4k\ln(1/\varepsilon)$ heads.

For the second claim, consider any $t \geq 8k\ln n + 6a\sqrt{k}\ln n$. Letting $\zeta_t = \sum_{i=1}^t Z_t$, the event $r_0 \geq t$ implies $\zeta_t < 4k\ln n = \frac{t}{2} - 3a\sqrt{k}\ln n = \mathbb{E}[\zeta_t] - \delta$ where $\delta = 3a\sqrt{k}\ln n$. By Hoeffding's inequality this event has probability at most $e^{-2\delta^2/t}$, and one can check that for all $a \geq 1$ we have $\frac{2\delta^2}{t} \geq a\ln n$.

# 4 Supplementary material for Section 7

## 4.1 Proof of Theorem 4

We state and prove two distinct theorems which immediately imply Theorem 4.

**Theorem 8.** *For all $0 < \gamma < 1/7$, all $d \geq 2$, and every (possibly randomized) learning algorithm, there exists an instance on $n \geq 2(\frac{1+\gamma}{8\gamma})^{\frac{d-1}{2}}$ points and $|\mathcal{C}| = 3$ latent clusters such that (1) all clusters have margin $\gamma$, and (2) to return with probability $2/3$ a clustering $\widehat{\mathcal{C}}$ such that $\triangle(\widehat{\mathcal{C}}, \mathcal{C}) = 0$ the algorithm must make $\Omega(n)$ same-cluster queries in expectation.*

*Proof.* The idea is the following. We define a single set of points $X \subset \mathbb{R}^d$ and randomize over the choice of the latent PSD matrix $W$; the claim of the theorem follows by applying Yao's minimax principle. Specifically, we let $X$ be a $\Theta(\sqrt{\gamma})$-packing of points on the unit sphere in $\mathbb{R}^d$. We show that, for $\boldsymbol{x} \in X$ drawn uniformly at random, setting $W = (1 + \gamma)\,\text{diag}(x_1^2, \ldots, x_d^2)$ makes $\boldsymbol{x}$ an outlier, as its distance $d_W(\boldsymbol{x}, \boldsymbol{0})$ from the origin is $1 + \gamma$, while every other point is at distance $\leq 1$. Since there are roughly $(1/\gamma)^d$ such points $\boldsymbol{x}$ in our set, the bound follows.

We start by defining the points $X$ in terms of their entry-wise squared vectors. Consider $S_d^+ = \mathbb{R}_+^d \cap S_d$ where $S_d = \{\boldsymbol{x} \in \mathbb{R}^d : \|\boldsymbol{x}\|_2 = 1\}$ is the unit sphere in $\mathbb{R}^d$. We want to show that there exists a set of $\frac{1}{2}(1/\varepsilon)^{d-1}$ points in $S_d^+$ whose pairwise distance is bigger than $\varepsilon/2$, where $\varepsilon$ will be defined later. To see this, recall that the packing number of the unit ball $B_d = \{\boldsymbol{x} \in \mathbb{R}^d : \|\boldsymbol{x}\|_2 \leq 1\}$ is $\mathcal{M}(B, \varepsilon) \geq (1/\varepsilon)^d$ —see, e.g., [6]. For $\varepsilon/2$ and $d - 1$, this implies there exists $Y \subseteq B_{d-1}$ such that $|Y| \geq (2/\varepsilon)^{d-1}$ and $\|\boldsymbol{y} - \boldsymbol{y}'\|_2 > \varepsilon/2$ for all distinct $\boldsymbol{y}, \boldsymbol{y}' \in Y$. Now, consider the lifting function $f : B_{d-1} \to \mathbb{R}^d$ defined by $f(\boldsymbol{y}) = (\sqrt{1 - \|\boldsymbol{y}\|_2^2}, y_1, \ldots, y_{d-1})$. Define the lifted set $Z = \{f(\boldsymbol{y}) : \boldsymbol{y} \in Y\}$. Clearly, every $\boldsymbol{z} \in Z$ satisfies $\|\boldsymbol{z}\|_2 = 1$ and $z_0 \geq 0$, so $\boldsymbol{z}$ lies on the northern hemisphere of the sphere $S_d$. Moreover, $\|f(\boldsymbol{y}) - f(\boldsymbol{y}')\|_2 \geq \|\boldsymbol{y} - \boldsymbol{y}'\|_2$ for any two $\boldsymbol{y}, \boldsymbol{y}' \in Y$. Hence, we have a set $Z$ of $(2/\varepsilon)^{d-1}$ points on the $d$-dimensional sphere such that $\|\boldsymbol{z} - \boldsymbol{z}'\|_2 > \varepsilon/2$ for all distinct $\boldsymbol{z}, \boldsymbol{z}' \in Z$. But a hemisphere is the union of $2^{d-1}$ orthants, hence some orthant contains at least $2^{-(d-1)}(2/\varepsilon)^{d-1} = (1/\varepsilon)^{d-1}$ of the points of $Z$. Without loss of generality we may assume this is the positive orthant and denote the set as $Z^+$.

We now define the input set $X \subseteq \mathbb{R}^d$ as follows:

$$X = X^+ \cup X^- = \{\sqrt{z} : z \in Z^+\} \cup \{-\sqrt{z} : z \in Z^+\}$$

Note that $n = |X| = 2|Z^+| = 2(1/\varepsilon)^{d-1}$. Next, we show how every $z \in Z^+$ defines a clustering instance satisfying the constraints of the thesis. For any $z^* \in Z^+$; let $w = (1+\gamma)z^*$ and $W = \operatorname{diag}(w_1, \ldots, w_d)$, which is PSD as required. Define the following three clusters:

$$C' = \{-\sqrt{z^*}\} \qquad C'' = \{\sqrt{z^*}\} \qquad C = X \setminus (C' \cup C'')$$

where, for $f : \mathbb{R} \to \mathbb{R}$, $f(x) = (f(x_1), \ldots, f(x_d))$. Since $C'$ and $C''$ are singletons, they trivially have weak margin $\gamma$. We now show that $C$ has weak margin $\gamma$ w.r.t. to $\mu = 0$; that is, $d_W(x, \mu)^2 > 1 + \gamma$ for $x = \pm\sqrt{z^*}$ and $d_W(x, \mu)^2 \leq 1$ otherwise. First, note that $d_W(x, \mu)^2 = \langle w, x^2 \rangle$. Now,

$$d_W(x, \mu)^2 = \begin{cases} (1+\gamma)\langle z^*, z^* \rangle = 1 + \gamma & \text{if } x \in C', C'' \\ (1+\gamma)\langle z^*, x^2 \rangle & \text{if } x \in C \end{cases} \tag{75}$$

However, by construction of $Z^+$, we have that for all $x \in C$ and $z = x^2$,

$$(\varepsilon/2)^2 \leq \|z - z^*\|_2^2 = \|z\|_2^2 - 2\langle z, z^* \rangle + \|z^*\|_2^2 = 2(1 - \langle z, z^* \rangle)$$

which implies $\langle z^*, x^2 \rangle \leq 1 - (\varepsilon/2)^2/2 = 1 - \varepsilon^2/8 = 1/(1+\gamma)$ for $\varepsilon = \sqrt{8\gamma/(1+\gamma)}$. Therefore (75) gives $d_W(x, \mu)^2 = (1+\gamma)\langle z^*, x^2 \rangle \leq 1$. This proves $C$ has weak margin $\gamma$ as desired.

The size of $X$ is:

$$n \geq 2\left(\frac{1}{\sqrt{8\gamma/(1+\gamma)}}\right)^{d-1} = 2\left(\frac{1+\gamma}{8\gamma}\right)^{\frac{d-1}{2}}$$

Now the distribution of the instances is defined by taking $z^*$ from the uniform distribution over $Z^+$. Consider any deterministic algorithm running over such a distribution. Note that same-cluster queries always return $+1$ unless at least one of the two queried points is not in $C$. As $C$ contains all points in $X$ but the symmetric pair $\sqrt{z^*}, -\sqrt{z^*}$ for a randomly drawn $z^*$, a constant fraction of the points in $X$ must be queried before one element of the pair is found with probability bounded away from zero. Thus, any deterministic algorithm that returns a zero-error clustering with probability at least $\delta$ for any constant $\delta > 0$ must perform $\Omega(n)$ queries. By Yao's principle for Monte Carlo algorithms then (see Section 1.4 above), any randomized algorithm that errs with probability at most $\frac{1-\delta}{2} \leq \frac{1}{2}$ for any constant $\delta > 0$ must make $\Omega(n)$ queries as well. $\square$

**Theorem 9.** *For all $\gamma > 0$, all $d \geq 48(1+\gamma)^2$, and every (possibly randomized) learning algorithm, there exists an instance on $n = \Omega\big(\exp(d/(1+\gamma)^2)\big)$ points and $|\mathcal{C}| = 2$ latent clusters such that (1) all clusters have margin at least $\gamma$, and (2) to return with probability $2/3$ a clustering $\widehat{\mathcal{C}}$ such that $\triangle(\widehat{\mathcal{C}}, \mathcal{C}) = 0$ the algorithm must make $\Omega(n)$ same-cluster queries in expectation.*

*Proof.* We exhibit a distribution of instances that gives a lower bound for every algorithm, and then use Yao's minimax principle. Let $p = \frac{1}{2(1+\gamma)}$. Consider a set of vectors $x_1, \ldots, x_n$ where every entry of each vector $x_{j,i}$ is i.i.d. and it is equal to 1 with probability. $p$. Define $X = \{x_1, \ldots, x_n\}$; note that in general $|X| \leq n$ since the points might not be all distinct. Let $x^\star = x_n$, $C = \{x_1, \ldots, x_{n-1}\}$, $C' = \{x^\star\}$. The latent clustering is $\mathcal{C} = \{C, C'\}$, and the matrix and center of $C$ are respectively $W = \operatorname{diag}(x^\star)$ and $c = 0$. The algorithms receive in input a random permutation of $X$; clearly, if it makes $o(|X|)$ queries, then it has vanishing probability to find $x^\star$, which is necessary to return the latent clustering $\mathcal{C}$.

Now we claim that, if $d \geq 48(1+\gamma)^2$, then we can set $n = \Omega\left(\exp\left(\frac{d}{48(1+\gamma)^2}\right)\right)$ and with constant probability we will have **(i)** $|X| = \Omega(n)$, and **(ii)** $C, C'$ have margin $\gamma$. This is sufficient, since the theorem then follows by applying Yao's minimax principle.

Let us first bound the probability that $|X| < n$. Note that for any two points $x_i, x_{i'}$ with $i \neq i'$ we have $\mathbb{P}(x_i = x_{i'}) = ((1-p)^2 + p^2)^d < (1 - \frac{1}{2(1+\gamma)})^d < e^{-\frac{d}{2(1+\gamma)}}$. Therefore, by a simple union bound over all pairs, $\mathbb{P}(|X| < n) < n^2 e^{-\frac{d}{2(1+\gamma)}}$.

Next, we want show that, loosely speaking, $d_W(\boldsymbol{x}, \boldsymbol{c})^2 \simeq dp$ for $\boldsymbol{x} \in C'$ whereas $d_W(\boldsymbol{x}, \boldsymbol{c})^2 \simeq dp^2$ for $\boldsymbol{x} \in C$; this will give the margin.

Now, for any $\boldsymbol{x}$,

$$d_W(\boldsymbol{x}, \boldsymbol{c})^2 = \sum_{i=1}^{d} x_i^\star (x_i - 0)^2 = \begin{cases} \sum_{i=1}^{d} x_i^\star x_i \sim B(d, p^2) & \boldsymbol{x} \in C \\ \sum_{i=1}^{d} x^\star \sim B(d, p) & \boldsymbol{x} \in C' \end{cases} \tag{76}$$

Where in the last equality we use the fact that the entries are unary, and where with the notation $B(d, p)$ we refer to a vector of length $d$ where each entry is equal to 1 with probability $p$. Let $\mu = dp^2$ and $\mu' = dp$, let $\varepsilon = 1/(1+\sqrt{2})$, and define

$$\phi = \mu(1 + \varepsilon), \qquad \phi' = \mu'(1 - \varepsilon\sqrt{p}) \tag{77}$$

By standard tail bounds,

$$\mathbb{P}(d_W(\boldsymbol{x}, \boldsymbol{c})^2 \geq \phi) \leq e^{-\frac{\varepsilon^2 \mu}{3}} \quad \text{for } \boldsymbol{x} \in C \tag{78}$$

$$\mathbb{P}(d_W(\boldsymbol{x}, \boldsymbol{c})^2 < \phi') < e^{-\frac{\varepsilon^2 p\mu'}{3}} = e^{-\frac{\varepsilon^2 \mu}{3}} \quad \text{for } \boldsymbol{x} \in C' \tag{79}$$

By a union bound on all points, the margin $\gamma_C$ of $C$ fails to satisfy the following inequality with probability at most $|X| e^{-\frac{\varepsilon^2 \mu}{3}} \leq n e^{-\frac{\varepsilon^2 \mu}{3}}$:

$$1 + \gamma_C = \frac{\min_{\boldsymbol{x} \notin C} d_W(\boldsymbol{x}, \boldsymbol{c})^2}{\max_{\boldsymbol{x} \in C} d_W(\boldsymbol{x}, \boldsymbol{c})^2} \geq \frac{\phi'}{\phi} = \frac{dp(1 - \varepsilon\sqrt{p})}{dp^2(1 + \varepsilon)} = \frac{1 - \varepsilon\sqrt{p}}{p(1 + \varepsilon)} \geq \frac{1}{2p} = 1 + \gamma \tag{80}$$

where the penultimate inequality holds since $\frac{1 - \varepsilon\sqrt{p}}{1 + \varepsilon} \geq \frac{1}{2}$ for our values of $p$ and $\varepsilon$. Note that, since $p = \frac{1}{2(1+\gamma)}$ and $n \leq \frac{1}{c} \exp\left(\frac{d}{48(1+\gamma)^2}\right) + 1$,

$$n e^{-\frac{\varepsilon^2 \mu}{3}} = n e^{-\frac{dp^2}{12}} = n e^{-\frac{d}{48(1+\gamma)^2}} \tag{81}$$

By one last union bound, the probability that $|X| = n$ and $\gamma_C \geq \gamma$ is at least

$$1 - n e^{-\frac{d}{48(1+\gamma)^2}} - n^2 e^{-\frac{d}{2(1+\gamma)}} \tag{82}$$

If $d \geq \frac{48}{(1+\gamma)^2}$, then we can let $n = \Omega\left(e^{\frac{d}{48(1+\gamma)^2}}\right)$ while ensuring the above probability is bounded away from 0.

The rest of the proof and the application of Yao's principle is essentially identical to the proof of Theorem 8 above. □

## 5 Comparison with SCQ-$k$-means

In this section we compare our algorithm to SCQ-$k$-means of [1]. We show that, in our setting, SCQ-$k$-means fails even on very simple instances, although it can still work under (restrictive) assumptions on $\gamma$, $W$, and the centers.

SCQ-$k$-means works as follows. First, the center of mass $\boldsymbol{\mu}_C$ of some cluster $C$ is estimated using $\mathcal{O}(\text{poly}(k, 1/\gamma))$ SCQ queries; second, all points in $X$ are sorted by their distance from $\boldsymbol{\mu}_C$ and the radius of $C$ is found via binary search. The binary search is done using same-cluster queries between the sorted points and any point already known to be in $C$. The margin condition ensures that, if we have an accurate enough estimate of $\boldsymbol{\mu}_C$, then the binary search will be successful (there are no inversions of the sorted points w.r.t. their cluster). This approach thus yields a $\mathcal{O}(\ln n)$ SCQ queries bound (the number of queries to estimate $\boldsymbol{\mu}_C$ is independent of $n$).

It is easy to see that this algorithm relies crucially on (1) each cluster $C$ must be spherical, and (2) the center of the sphere must coincide with the centroid $\boldsymbol{\mu}_C$. In formal terms, the setting of [1] is a special cases of ours where for all $C$ we have $W_C = I_d$ and $\boldsymbol{c} = \mathbb{E}_{\boldsymbol{x} \in C}[\boldsymbol{x}]$. If any of these two assumptions does not hold, then it is easy to construct instances where [1] fails to recover the clusters and, in fact, achieves error very close to a completely random labeling. Formally:

**Lemma 10.** *For any fixed $d \geq 2$, any $p \in (0, 1)$, and any sufficiently small $\gamma > 0$, there are arbitrarily large instances on $n$ points and $k = 2$ clusters on which SCQ-$k$-means incurs error $\triangle(\widehat{\mathcal{C}}, \mathcal{C}) \geq \frac{1-p}{2}$ with probability at least $1 - p$.*

*Sketch of the proof.* We describe the generic instance on $n$ points for $d = 2$. The latent clustering $\mathcal{C}$ is formed by two clusters $C_1, C_2$ of size respectively $n_1 = n\frac{1+p}{2}$ and $n_2 = n\frac{1-p}{2}$. In $C_1$, half of the points are in $(1, 0)$ and half in $(-1, 0)$. In $C_2$, all points are in $(0, \frac{\sqrt{1+\gamma}}{2})$. (One can in fact perturb the instance so that all points are distinct without impairing the proof). For both clusters, the center coincide with their center of mass, $\boldsymbol{\mu}_1 = (0, 0)$ and $\boldsymbol{\mu}_2 = (0, \frac{\sqrt{1+\gamma}}{2})$. For both clusters, the latent metric is given by the PSD matrix $W = \left(\begin{smallmatrix} .25 & 0 \\ 0 & 1 \end{smallmatrix}\right)$. It is easy to see that $d_W(\boldsymbol{x}, \boldsymbol{\mu}_1)^2 = 1/4$ if $\boldsymbol{x} \in C_1$ and $d_W(\boldsymbol{x}, \boldsymbol{\mu}_1)^2 = (1+\gamma)/4$ if $\boldsymbol{x} \in C_2$, and so $C_1$ has margin exactly $\gamma$. On the other hand $C_2$ has margin $\gamma$ since $d_W(\boldsymbol{x}, \boldsymbol{\mu}_2)^2 = 0$ if $\boldsymbol{x} \in C_2$ and $d_W(\boldsymbol{x}, \boldsymbol{\mu}_2)^2 > 0$ otherwise.

Figure 4: A bad instance for SCQ-$k$-means. With good probability, the algorithm classifies all points in a single cluster, incurring error $\simeq 1/2$, the same as a random labeling.

Now consider SCQ-$k$-means. The algorithm starts by sampling at least $\frac{k \ln(k)}{\gamma^4}$ points from $X$ and setting $\widehat{\boldsymbol{\mu}}$ to the average of the points with the majority label. By standard concentration bounds then, for $\gamma$ small enough, with probability at least $1 - p$ the majority cluster will be $C_1$ and the estimate $\widehat{\boldsymbol{\mu}}$ of its center of mass $(0, 0)$ will be sufficiently close to $\boldsymbol{\mu}_1$ that the ordering of all points in $X$ by their Euclidean distance w.r.t. $\widehat{\boldsymbol{\mu}}$ will set all of $C_2$ before all of $C_1$. But since $n_2 = n\frac{1-p}{2}$, the median of the sorted sequence will be a point of $C_1$. Thus the binary search will make its first query on a point of $C_1$ and will continue thereafter classifying all of $X$ as belonging to $C_1$. Thus the algorithm will output the clustering $\widehat{\mathcal{C}} = \{X, \emptyset\}$ which gives $\triangle(\widehat{\mathcal{C}}, \mathcal{C}) = \frac{1-p}{2}$. $\qquad\square$

Next, we show that the approach [1] still works if one relaxes the assumption $W = I$, at the price of strengthening the margin $\gamma$. Let $\lambda_{\max}$ and $\lambda_{\min} > 0$ be, respectively, the largest and smallest eigenvalues of $W$. The condition number $\kappa_W$ of $W$ is the ratio $\lambda_{\max}/\lambda_{\min}$. If $\kappa_W$ is not too large, then $W$ does not significantly alter the Euclidean metric, and the ordering of the points is preserved. Formally:

**Lemma 11.** *Let $\kappa_W$ be the condition number of $W$. If every cluster $C$ has margin at least $\kappa_W - 1$ with respect to its center of mass $\boldsymbol{\mu}_C$, and if we know $\boldsymbol{\mu}_C$, then we can recover $C$ with $\mathcal{O}(\ln n)$ SCQ queries.*

*Proof.* Fix any cluster $C$ and let $\boldsymbol{\mu} = \boldsymbol{\mu}_C$. For any $\boldsymbol{z} \in \mathbb{R}^d$ we have $\lambda_{\min}\|\boldsymbol{z}\|_2^2 \leq \|\boldsymbol{z}\|_W^2 \leq \lambda_{\max}\|\boldsymbol{z}\|_2^2$ where $\lambda_{\min}$ and $\lambda_{\max}$ are, respectively, the smallest and largest eigenvalue of $W$. Sort all other points $\boldsymbol{x}$ by their Euclidean distance $\|\boldsymbol{x} - \boldsymbol{\mu}\|_2$ from $\boldsymbol{\mu}$. Then, for any $\boldsymbol{x} \in C$ and any $\boldsymbol{y} \notin C$ we have:

$$\frac{\|\boldsymbol{y} - \boldsymbol{\mu}\|_2^2}{\|\boldsymbol{x} - \boldsymbol{\mu}\|_2^2} \geq \frac{\lambda_{\min}}{\lambda_{\max}} \frac{\|\boldsymbol{y} - \boldsymbol{\mu}\|_W^2}{\|\boldsymbol{x} - \boldsymbol{\mu}\|_W^2} = \frac{1}{\kappa_W} \frac{d(\boldsymbol{y}, \boldsymbol{\mu})^2}{d(\boldsymbol{x}, \boldsymbol{\mu})^2} > \frac{1 + \gamma}{\kappa_W} \tag{83}$$

Hence, if $\gamma \geq \kappa_W - 1$, there is $r \geq 0$ such that $\|\boldsymbol{x} - \boldsymbol{\mu}\|_2 \leq r$ for all $\boldsymbol{x} \in C$ and $\|\boldsymbol{y} - \boldsymbol{\mu}\|_2 \geq r$ all $\boldsymbol{y} \notin C$. We can thus recover $C$ via binary search as in [1]. $\qquad\square$

As a final remark, we observe that the above approach is rather brittle, since $\kappa_W$ is unknown (because $W$ is), and if the condition $\kappa_W \leq 1 + \gamma$ fails, then once again the binary search can return a clustering far from the correct one.

# 6 Comparison with metric learning

In this section we show that metric learning, a common approach to latent cluster recovery and related problems, does not solve our problem even when combined with same-cluster and comparison queries. Intuitively, we want to learn an approximate distance $\widehat{d}$ that preserves the ordering of the distances between the points. That is, for all $\boldsymbol{x}, \boldsymbol{y}, \boldsymbol{z} \in X$, $d(\boldsymbol{x}, \boldsymbol{y}) \leq d(\boldsymbol{x}, \boldsymbol{z})$ implies $\widehat{d}(\boldsymbol{x}, \boldsymbol{y}) \leq \widehat{d}(\boldsymbol{x}, \boldsymbol{z})$. If this holds then $d$ and $\widehat{d}$ are equivalent from the point of view of binary search. To simplify the task, we may equip the algorithm with an additional *comparison query* CMP, which takes in input two pairs of points $\boldsymbol{x}, \boldsymbol{x}'$ and $\boldsymbol{y}, \boldsymbol{y}'$ from $X$ and tells precisely whether $d(\boldsymbol{x}, \boldsymbol{x}') \leq d(\boldsymbol{y}, \boldsymbol{y}')$ or not. It turns out that, even with SCQ+CMP queries, learning such a $\widehat{d}$ requires to query essentially all the input points.

**Theorem 10.** *For any $d \geq 3$, learning any $\widehat{d}$ such that, for all $\boldsymbol{x}, \boldsymbol{y}, \boldsymbol{z} \in X$, if $d(\boldsymbol{x}, \boldsymbol{y}) \leq d(\boldsymbol{x}, \boldsymbol{z})$ then $\widehat{d}(\boldsymbol{x}, \boldsymbol{y}) \leq \widehat{d}(\boldsymbol{x}, \boldsymbol{z})$, requires $\Omega(n)$ SCQ+CMP queries in the worst case, even with an arbitrarily large margin $\gamma$.*

*Proof.* We reduce the problem of learning the order of pairwise distances induced by $W$, which we call ORD, to the problem of learning a separator hyperplane, which we call SEP and whose query complexity is linear in $n$.

Problem SEP is as follows. The inputs are a set $X = \{\boldsymbol{x}_1, \ldots, \boldsymbol{x}_n\} \subset \mathbb{R}^d$ (the observations) and a set $\mathcal{H} = \{\boldsymbol{h}_1, \ldots, \boldsymbol{h}_k\} \subset \mathbb{R}_+^d$ (the hypotheses). We require that $\boldsymbol{h}_j \in \mathbb{R}_+^d$. We have oracle access to $\sigma : X \to \{+1, -1\}$ such that $\sigma(\cdot) = \operatorname{sgn}\langle \boldsymbol{h}, \cdot \rangle$ for some $\boldsymbol{h} \in \mathcal{H}$. The output is the $\boldsymbol{h} \in \mathcal{H}$ that agrees with $\sigma$. We assume $\mathcal{H}, X$ support a margin: $\exists \varepsilon > 0$, possibly dependent on the instance, such that $\operatorname{sgn}\langle \boldsymbol{h}, \boldsymbol{x} \rangle = \operatorname{sgn}\langle \boldsymbol{h}, \boldsymbol{x}' \rangle$ for all $\boldsymbol{x}'$ with $\|\boldsymbol{x} - \boldsymbol{x}'\| \leq \varepsilon$. (Note that this is *not* the cluster margin $\gamma$).

Let $Q_{\text{ORD}}(n)$ and $Q_{\text{SEP}}(n)$ be the query complexities of ORD and SEP on $n$ points. We show:

**Lemma 12.** $Q_{ORD}(3n) \leq Q_{SEP}(n)$.

*Proof.* Let $X = \{\boldsymbol{x}_1, \ldots, \boldsymbol{x}_n\} \subseteq \mathbb{R}^d$ be the input points for SEP and let $\boldsymbol{h} \in \mathbb{R}_+^d$ be the target hypothesis. By scaling the dataset we can assume $\|\boldsymbol{x}_i\| \leq \varepsilon$ for any desired $\varepsilon$ (even dependent on $n$). We define an instance of ORD on $n' = 3n$ points as follows. First, $W = \operatorname{diag}(\boldsymbol{h})$. Second, the input set is $X' = S_1 \cup \ldots \cup S_n$ where for $i = 1, \ldots, n$ we define $S_i = \{\boldsymbol{a}_i, \boldsymbol{b}_i, \boldsymbol{c}_i\}$ with:

$$\boldsymbol{a}_i = 6^i \cdot \mathbf{1} \tag{84}$$
$$\boldsymbol{b}_i = 2 \cdot \boldsymbol{a}_i \tag{85}$$
$$\boldsymbol{c}_i = 3 \cdot \boldsymbol{a}_i + \boldsymbol{x}_i \tag{86}$$

We first show that a solution to ORD gives a solution of SEP. Suppose indeed that for all pairs of points $\{\boldsymbol{q}, \boldsymbol{p}\}, \{\boldsymbol{x}, \boldsymbol{y}\}$ we know whether $d_W(\boldsymbol{q}, \boldsymbol{p}) \leq d_W(\boldsymbol{x}, \boldsymbol{y})$. This is equivalent to knowing the output of CMP$(\{\boldsymbol{q}, \boldsymbol{p}\}, \{\boldsymbol{x}, \boldsymbol{y}\})$, which is

$$\text{CMP}(\{\boldsymbol{q}, \boldsymbol{p}\}, \{\boldsymbol{x}, \boldsymbol{y}\}) = \operatorname{sgn}\left\langle \boldsymbol{h}, (\boldsymbol{q} - \boldsymbol{p})^2 - (\boldsymbol{x} - \boldsymbol{y})^2 \right\rangle \tag{87}$$

Consider then the point $\boldsymbol{q} = \boldsymbol{c}_i, \boldsymbol{p} = \boldsymbol{x} = \boldsymbol{b}_i, \boldsymbol{y} = \boldsymbol{a}_i$ for each $i$. Then:

$$\text{CMP}(\{\boldsymbol{q}, \boldsymbol{p}\}, \{\boldsymbol{x}, \boldsymbol{y}\}) = \operatorname{sgn}\left\langle \boldsymbol{h}, (\boldsymbol{a}_i - \boldsymbol{b}_i)^2 - (\boldsymbol{b}_i - \boldsymbol{c}_i)^2 \right\rangle \tag{88}$$
$$= \operatorname{sgn}\left\langle \boldsymbol{h}, (\boldsymbol{a}_i)^2 - (-\boldsymbol{a}_i - \boldsymbol{x}_i)^2 \right\rangle \tag{89}$$
$$= \operatorname{sgn}\left\langle \boldsymbol{h}, 2 \cdot 6^i \boldsymbol{x}_i - \boldsymbol{x}_i^2 \right\rangle \tag{90}$$
$$= \operatorname{sgn}\left\langle \boldsymbol{h}, \boldsymbol{x}_i \left(1 - \frac{\boldsymbol{x}_i}{2 \cdot 6^i}\right) \right\rangle \tag{91}$$

By the margin hypothesis, for $\varepsilon$ small enough this equals $\operatorname{sgn}(\langle \boldsymbol{h}, \boldsymbol{x}_i \rangle)$, i.e., the label of $\boldsymbol{x}_i$ in SEP.

We now show that all the other queries reveal no information about the solution of SEP. Suppose then the points are not in the form $\boldsymbol{q} = \boldsymbol{c}_i, \boldsymbol{p} = \boldsymbol{x} = \boldsymbol{b}_i, \boldsymbol{y} = \boldsymbol{a}_i$. Without loss of generality, we can assume that $\boldsymbol{q} > \boldsymbol{p}$ and $\boldsymbol{q} \geq \boldsymbol{x} > \boldsymbol{y}$. It is then easy to see that, for $\varepsilon$ small enough, $(\boldsymbol{q} - \boldsymbol{p})^2 - (\boldsymbol{x} - \boldsymbol{y})^2 > 0$ or $(\boldsymbol{q} - \boldsymbol{p})^2 - (\boldsymbol{x} - \boldsymbol{y})^2 < 0$. This holds independently of the $\boldsymbol{x}_i$ and of $W$ and therefore gives no information about the solution of SEP.

It follows that, if we can solve ORD in $f(3n)$ CMP queries, then we can solve SEP in $f(n)$ queries. Finally, note that adding SCQ queries does not reduce the query complexity (e.g., let $X$ lie in a single cluster). For the same reason, we can even assume an arbitrarily large cluster margin $\gamma$. $\qquad \square$

It remains to show that SEP requires $\Omega(n)$ CMP queries in the worst case. This is well known, but we need to ensure that $\mathcal{H} \subset \mathbb{R}^d_+$ and that any $h \in \mathcal{H}$ supports a margin as described above.

Consider the following set $X = \{\boldsymbol{x}_1, \ldots, \boldsymbol{x}_n\} \subseteq \mathbb{R}^3$:

$$\boldsymbol{x}_i = (1 - \delta, -\cos(\theta_i), -\sin(\theta_i)) \tag{92}$$

where $\theta_i = i\frac{\pi}{2n}$ and $\delta$ is sufficiently small. Let $\mathcal{H} = \{\boldsymbol{h}_1, \ldots, \boldsymbol{h}_n\}$, where

$$\boldsymbol{h}_j = (1, \cos(\theta_j), \sin(\theta_j)) \tag{93}$$

Note that $\mathcal{H} \subset \mathbb{R}^d_+$ as required. Clearly:

$$\langle \boldsymbol{h}_j, \boldsymbol{x}_i \rangle = \begin{cases} -\delta & \text{if } j = i \\ 1 - (\delta + \cos(\theta_i - \theta_j)) & \text{if } j \neq i \end{cases} \tag{94}$$

By choosing $\delta = \frac{1 - \cos(\pi/2n)}{2}$ we have sgn $\langle \boldsymbol{h}, \boldsymbol{x}_i \rangle = -1$ if and only if $i = j$. Clearly, any algorithm needs to probe $\Omega(n)$ labels to learn $h$ with constant probability for some $h \in \mathcal{H}$. Finally, note that any $h$ supports a margin, as required. $\square$