[Reviews · NeurIPS 2020]

Review 1

Summary and Contributions: This work addressees the problem of recovering the cluster structure when the clusters are not necessarily spherical. The paper uses the framework of semi-supervised active clustering where the algorithm has access to a same-cluster oracle (which can answer whether two points belong to the same or different clusters). Their main contribution is providing theoretical guarantees (on the run-time and the number of queries) when the underlying spherical cluster has been transformed by a series of rotations and scalings, in other words, is ellipsoidal.

Strengths: The main technical innovation of the work comes from the use of MVEE (or a minimum volume enclosing ellipsoid). The main idea can be summarized as follows. 1. Get enough points from a cluster. 2. Bound it using an mvee. Using classical results, we know that the mvee contains atleast half of the points of the cluster. 3. This set can be covered using a set of uniform (referred to as mono-chromatic in the paper) hyper-rectangles. I believe this is the most important technical contribution of this work and is novel.

Weaknesses: One of the main weaknesses of the work comes from the upper bounds on the number of queries which is exponential in r = max rank(C). The authors make a comment that this is "desirable as real-world data often exhibits a low-rank" (r << d). In the context of applications like matrix factorization this makes sense as it is reasonable to assume that the number of latent variables are small so r << d. What I am having a bit of trouble understanding is how we can justify the assumption that a cluster has low rank (and subsequently all clusters have low rank). In the context of a cluster, does this imply that all the points of a cluster can be represented as a linear combination of a small number (r) of points in that cluster?

Correctness: While I did not read the detailed proof in the supplementary, the description in the main version was adequate to get an intuition of the proof. And I think that the results claimed are true.

Clarity: The paper is well written.

Relation to Prior Work: Prior work is properly discussed.

Reproducibility: Yes

Additional Feedback: The paper looks on the gamma margin notion. It might also be interesting to apply the same line of analysis for other niceness notions (for example alpha-center proximity).


Review 2

Summary and Contributions: This paper studies the exact recovery of ellipsoidal clusters with same cluster queries. This is done by relaxing the \gamma-margin notion in the center-based clustering. This relaxation overcomes the problem of the strong assumption in k-means that the clusters are spherical. The author proposes an algorithm that uses O(k^3 ln(k) ln(n)) queries, and O(kn +k^3) running time. This notation hides the exponential dependence on the dimension of the clustering and the margin. The authors show lower bound that shows this dependency is necessary.

Strengths: I agree that in this specific setting of the problem (same-cluster queries), the assumption discuss in the above section is unrealistic. Hence, I was convinced that this problem is indeed interesting both practically and theoretically. I believe this paper will lead to an interesting follow-up works on this problem and other related problems. The authors show a novel technique named monochromatic tessellation. It exploits the margin definition to divide the calculated ellipsoid into monochromatic subsets. I found this technique elegant and explained very interestingly. I found the extension of the margin notion simple and elegant.

Weaknesses: In the experiments, the authors generated synthetics instances (ellipsoids) which suit explicitly to the setting they studied, while not suited to the setting of [4]( k-means). Hence, the results are not surprising. I believe running the experiments on real-world data sets would be more engaging and will help to motivate this algorithm for practitioners.

Correctness: All the proofs are found in the appendix. I only briefly viewed the claims, which make sense. However, I did not completely verify them.

Clarity: This paper is written very well, I enjoyed reading it.

Relation to Prior Work: Yes.

Reproducibility: Yes

Additional Feedback: - Do the authors use the construction in theorem 3 (Low-stretch separators) in the final algorithm in the experiments? -row 327-330 - I did not understand how does the heuristic work. - - - - - - - Edit: I have read the author's response. It strengthened my opinion that this paper should be accepted, especially the response to reviewer 5.


Review 3

Summary and Contributions: This paper studies the SSAC setting introduced by Ashtiani et.al. and extends its applications significantly. In this paper, the clusters can be ellipsoidal (instead of spherical) and the authors design an efficient and elegant algorithm to recover the clusters in this significantly more general setting.

Strengths: The contribution is novel, interesting, and extends the applicability of Ashtiani's algorithms significantly.

Weaknesses: The writing can be improved a bit although it is difficult given the lack of space. I would like to see a definition of MVEE in the beginning. Further, eq (1) is not clear to me since the definition of E/d is ambiguous.

Correctness: The claims and methods are correct.

Clarity: The paper is well written but can be improved

Relation to Prior Work: Yes

Reproducibility: Yes

Additional Feedback: Minor comments: 1) Please insert references for this statement: This captures k-means, k-medians, k-centers, their variants, and any scenario where features have been arbitrarily scaled, or a portion of the cluster is unavailable due to errors or privacy reasons. 2) A few real-world experiments on the lines of https://arxiv.org/abs/1806.05938 would have been more interesting and further strengthened this paper. After the Rebuttal I have read the rebuttal and I have no additional comments


Review 4

Summary and Contributions: This work considers the problem of clustering using same-cluster oracle queries. One of the main differences from prior work on objective based clustering is that the paper considers clusters that can be ellipsoid in shape.

Strengths: 1. The sub-linear complexity under certain assumptions is quite interesting. 2. The paper presents lower and upper bounds for the problem.

Weaknesses: 1. Theorem 1 is difficult to follow in intro without the notation. I think authors should add intuition or a simpler statement in the intro. 2. What happens when W is identity matrix? Then the problem reduces spherical case and how does the results compare with prior work. 3. Experiments are weak. (More details below.)

Correctness: The claims are correct but the empirical evaluation is on synthetic data and can be improved.

Clarity: Initial problem definition is not very clear.

Relation to Prior Work: No, the authors build on [4] even though [3] got rid of the assumptions of [4].

Reproducibility: Yes

Additional Feedback: Post Rebuttal: Authors have addressed all the questions about the theoretical contributions. I encourage authors to strengthen the experiments with real data and baselines. -------------- 1. The Preliminaries section just mentions that the clustering C is consistent with W. W is just defining the distance between every pair of points. Assuming it induces a seminorm does not decide the optimization objective to construct C. Are we considering k-means objective under d_W metric? Or are we assuming that the clusters can be arbitrary but the points form ellipsoid according to a W that satisfies gamma margin? - I am guessing it is the second case. Please clarify if I am correct. Right now it is very confusing because authors present introduction wrt k-means objective but then problem definition deals with any ground truth clustering that is ellipsoid. A compelling argument could be that the ground truth clustering techniques require \Omega(n) queries to identify the clusters. This work considers a setting where the points are present in a euclidean space and the clusters have gamma margin. Under this assumption, the proposed technique achieves sublinear complexity. Also please add a formal problem definition in section 2. 2. Do you assume that the cluster size is equal or \Theta(n)? 3. Right now the algorithm assumes gamma is known apriori. What if gamma is not known apriori? Prior work assumes that the data satisfies gamma margin for some gamma but do not require its estimate. 4. The analysis crucially depends on the dimension d being a constant. It would be nice to add a result when dimension is say log n. 5. Experiments are weak. The datasets considered are all synthetic. It is difficult to validate the effectiveness on such datasets. Better mechanism to generate data: Take a real dataset. Consider clusters according to k-clustering or optimal clusters. Now perturb the clusters by scaling the distance and changing the cluster shape to elliptical instead of sphere. Compare baselines on the original and scaled version to demonstrate the effectiveness. It would be interesting to see the change in performance by changing W (unit matrix to ellipsoid). The baselines considered are very suboptimal and not suited for the setting considered. Since the clusters generated are not based on any objective optimization, authors should compare with techniques that consider ground truth clustering (irrespective of shape) [30]. 6. Authors motivate the problem with k-means citing [4] but the reference [3] is an improvement that does not assume gamma margin in their formulation. Authors should consider [3] for all purposes. Overall, I like the algorithmic contribution and ideas used to solve the problem but the paper is not written clearly and empirical evaluation is quite weak.

[Author Response · NeurIPS 2020]

We thank all reviewers for their work. We ask Reviewer #5 to reconsider his/her decision in the light of our feedback.

**About our experiments.** We agree that experiments on real-world datasets would add value. However, we remark
that the main objective of our work is a theoretical investigation of the query complexity of cluster recovery with
same-cluster queries. Performing a thorough evaluation on real-world data is a separate task, beyond the scope of this
paper. The goal of our experiments is to certify that our algorithms are easy to implement, behave as predicted by the
theory, and have no large hidden costs. We believe our empirical setting fulfills these objectives.

**Reviewer #2.**
*Observation about low rank*: The statement about linear combinations is correct and is equivalent to a low-rank
assumption. A justification is given precisely by matrix factorization, which assumes the set of users (the rows of the
user matrix) spans a low-dimensional subspace. Another justification is that often some features (dimensions) are linear
combinations of others, and the actual clusters lie in a low dimensional subspace (namely, are contained in degenerate
ellipsoids). In any case, we note that the exponential dependence on $r$ is unavoidable, as our lower bounds show.
*Other niceness notions*: we agree on these future directions, and thank the reviewer for the suggestion.

**Reviewer #3.**
*Low-stretch separators*: no, we do not use them — they are a "bonus" theoretical contribution.
*The heuristic*: we agree and will clarify this.

**Reviewer #4.**
*MVEE and Eq. 1*: we agree and will clarify this.
*References for $k$-center etc.*: we agree and will add them.
*Datasets*: we thank the reviewer for the pointer.

**Reviewer #5.**
*"Theorem 1 is difficult to follow" and "What happens when $W = I$?"*: we believe these comments do not identify
"typical weaknesses", since they just concern the way a theorem is introduced and a special case of our results.
*Theorem 1 is difficult to follow*: we will add more intuition and simplify the statement of the theorem.
*What happens when $W = I$?*: our algorithms are not adaptive in this sense, so when $W = I$ they do not obey the
bounds of [4]. We agree this is an interesting direction. Note however that, in the supplementary material, we show that
the bounds of [4] hold when $W$ has low condition number and thus is "close" to $I$.
*We should compare to [3] instead of [4]:* we disagree. The goal of [3] is to build a PTAS for $k$-means. Our goal is to
recover a latent clustering. The two problems are incomparable: a good $k$-means value can be achieved by a clustering
very different from the optimal one, and vice versa. And indeed, [3] does not require any margin, whereas our lower
bounds clearly say that without margin one needs $n$ queries. This would have implied a contradiction had the two works
solved the same problem. Thus, [4] provides a baseline, but [3] does not.
*Problem formulation*: thanks for the suggestion. We will add a formal definition of the problem addressed in the paper.
*Balanced clusters*: we do not need any assumption on the cluster sizes. Since there are at most $k$ clusters, there is
always a cluster with $\Omega(n/k)$ points, and this is sufficient for our algorithm.
*Knowledge of $\gamma$*: we disagree. Our algorithms require knowledge of $\gamma$ exactly as the algorithms of [4] do: if the value
of $\gamma$ passed to the algorithm is a lower bound on the actual margin of the instance, then the correctness is guaranteed,
otherwise the algorithm is allowed to return an arbitrary clustering. We are not aware of prior work that does not require
knowing $\gamma$ for exact recovery.
*The analysis crucially relies on $d$ being a constant*: this is not true. Our analysis does not make any assumption on $d$,
and indeed $d$ appears in our bounds. If the reviewer refers to the bounds in the abstract, then, as the abstract says, those
are simplified versions of the full bounds.
*We should experimentally compare to [30]*: we disagree, [30] requires in input a similarity matrix $W$ correlated to the
latent clustering, which is not available in our setting.
*The paper is not written clearly*: we will do our best to clarify all definitions, assumptions, and main results earlier on in
the paper.

[Meta-Review · NeurIPS 2020]

The paper makes a strong theoretical contribution. The experiments can be improved since they are only on synthetic data (which is also nice for the algorithm presented). However, the theoretical results carry the paper.